# Specific length and structure rather than high thermodynamic stability enable regulatory mRNA stem-loops to pause translation

Chen Bao [1], Mingyi Zhu [1], Inna Nykonchuk[1], Hironao Wakabayashi[1], David H. Mathews [1] &
Dmitri N. Ermolenko [1✉]

Translating ribosomes unwind mRNA secondary structures by three basepairs each elongation cycle. Despite the ribosome helicase, certain mRNA stem-loops stimulate programmed ribosomal frameshift by inhibiting translation elongation. Here, using mutagenesis, biochemical and single-molecule experiments, we examine whether high stability of three basepairs, which are unwound by the translating ribosome, is critical for inducing ribosome pauses. We find that encountering frameshift-inducing mRNA stem-loops from the *E. coli dnaX* mRNA and the *gag-pol* transcript of Human Immunodeficiency Virus (HIV) hinders A-site tRNA binding and slows down ribosome translocation by 15-20 folds. By contrast, unwinding of first three basepairs adjacent to the mRNA entry channel slows down the translating ribosome by only 2-3 folds. Rather than high thermodynamic stability, specific length and structure enable regulatory mRNA stem-loops to stall translation by forming inhibitory interactions with the ribosome. Our data provide the basis for rationalizing transcriptome-wide studies of translation and searching for novel regulatory mRNA stem-loops.

---

[1] Department of Biochemistry & Biophysics at School of Medicine and Dentistry and Center for RNA Biology, University of Rochester, Rochester, NY, USA.
✉email: dmitri_ermolenko@urmc.rochester.edu

During translation elongation, the ribosome translocates along mRNA one codon at a time. The narrow mRNA channel of the small ribosomal subunit[1–3] accommodates mRNA only when it is not basepaired. Hence, translation requires unwinding of mRNA secondary structure. Indeed, the translating ribosome has been demonstrated to be an efficient helicase[4,5] that unwinds three basepairs per translocation step. However, it remains unclear to what extent mRNA stem-loops reduce the rate of translation elongation relative to translation along unpaired mRNA. Because mRNAs have a propensity to form intramolecular basepairing interactions and were shown to fold into extensive secondary structures in live cells[6–9], elucidating the effects of mRNA stem-loops on translation elongation has important implications for translation regulation.

Several published studies vary in their estimates for the magnitude of translation elongation slowdown induced by mRNA secondary structures[5,10–13]. For instance, optical tweezer experiments suggested that ribosome translocation through three consecutive GC basepairs is only 2–3-fold slower than translocation along unpaired RNA[10,12]. Consistent with these data, several transcriptome-wide ribosome profiling analyses performed in yeast and *E. coli* cells indicated that most of the secondary structure elements within coding regions of mRNA do not produce detectable ribosome pauses or significantly change translational efficiency[14,15].

In contrast to aforementioned reports demonstrating modest impact of mRNA secondary structure on translation rate, a ribosome profiling study performed in *E. coli* revealed strong negative correlation between translational efficiency and the presence of extensive intramolecular basepairing interactions in ORFs suggesting that both initiation and elongation phases of protein synthesis can be regulated by mRNA secondary structure[16]. Another study suggested that the effects of mRNA secondary structure on translational efficiency measured by ribosome profiling may be masked by the presence of codons that are decoded by abundant tRNAs in the structured regions of ORFs[17]. These studies indicate that mRNA secondary structure may strongly affect the translation elongation rate.

In addition to adjusting the translation elongation rate, certain RNA stem-loop structures were found to induce ribosome stalling that results in accumulation of truncated polypeptides[18] and No-Go mRNA decay[19]. Furthermore, certain mRNA stem-loops and pseudoknots are known to trigger programmed translational pauses[20] and stimulate −1 programmed ribosomal frameshifting (PRF) by slowing down the ribosome[21]. −1 PRF plays an important role in regulating synthesis of bacterial, viral and eukaryotic proteins, including translation of gag-pol polyprotein of human immunodeficiency virus (HIV)[22] and C-terminally extended polyprotein in SARS-CoV-2 coronavirus, which caused the COVID-19 pandemic[23,24].

−1 PRF requires the presence of two signals in an mRNA: the heptanucleotide slippery sequence NNNWWWN (where N can be any nucleotide and W is either A or U) and a downstream frameshift-inducing hairpin or a pseudoknot called the frameshift stimulating sequence (FSS)[25]. The slippery sequence allows cognate or near-cognate pairing of the P-site and A-site tRNAs in both 0 and −1 frames and thus makes frameshift thermodynamically favorable[26]. A number of previous studies demonstrated that when the slippery sequence is bound in the P and A sites of the ribosome and the FSS is positioned at the entry to mRNA tunnel of the small subunit, the rate of tRNA/mRNA translocation decreases by approximately an order of magnitude[11,27–31]. We also recently demonstrated that, when positioned 11–12 nucleotides downstream of the P-site codon, FSSs can transiently escape the ribosome helicase and inhibit A-site tRNA binding by docking into the A site of the small (30S)

ribosomal subunit[31]. Hence, the FSS slows down the ribosome through two alternative pathways: inhibiting translocation and A-site tRNA binding.

It is commonly assumed that mRNA stem-loops and pseudoknots induce −1 PRF and ribosome pausing due to their high overall thermodynamic stability[21,32,33]. However, the translating ribosome unwinds only three basepairs at a time and thus is unable to "sense" the overall thermodynamic stability of mRNA stem-loops or pseudoknots. Consistent with this idea, optical tweezer experiments demonstrated that when GC content remains constant throughout very long stem-loops, the ribosome can unwind dozens basepairs with a seemingly uniform rate[5]. It has also been hypothesized that rather than the total stability of the FSS, high local thermodynamic stability of basepairs positioned at the entrance to the mRNA tunnel of the small subunit that are first unwound by the ribosome explain the FSS-induced ribosome pausing[32,34,35]. Finding why some stem-loops slow down the ribosome more than others is important for interpreting ribosome-profiling data, discovering new regulatory stem-loops and designing mRNAs for biotechnology (including mRNA vaccines).

Here, we investigate to what degree high thermodynamic stability of the three basepairs, which are unwound by the translating ribosome, accounts for translation pausing mediated by frameshift-inducing mRNA stem-loops. We hypothesize that in addition to overall and local thermodynamic stability, other structural properties enable stem-loops to induce ribosome pausing. To test these hypotheses, we examine how sequence and structure perturbations of frameshift-inducing stem-loops affect their ability to slow down translation elongation.

We find that unwinding of three basepairs per elongation cycle only moderately affects the rate of translation elongation. By contrast, when encountered by the translating ribosome, frameshift-inducing stem-loops produce an order-of-magnitude longer translational pause. Our results indicate that specific length and the presence of the loop are important for inducing long ribosome pauses. Hence, our findings offer new insights into translational control by mRNA secondary structures and provide rationale for the discovery of previously unknown regulatory secondary structure elements at transcriptome-wide level.

## Results

We used a combination of smFRET and biochemical assays to investigate properties of frameshift-inducing stem-loops from *E. coli dnaX* mRNA and the *gag-pol* transcript of HIV that are important for inducing ribosome pausing and −1 PRF. The *dnaX* mRNA encodes the τ and γ subunits of DNA polymerase III[36]. The slippery sequence AAAAAAG and a downstream 11 basepair-long FSS mRNA hairpin, which are separated by a five nucleotide-long spacer, program −1 PRF that produces γ subunit with 50–80% efficiency[29,32,37] (Fig. 1). While the Gag polyprotein containing HIV structural proteins is produced in the absence of frameshifting, the 12 basepair-long RNA hairpin in combination with the slippery sequence UUUUUUA induces −1 PRF with 5–10% efficiency[22,34]. This frameshifting yields the Gag-Pol polyprotein, which contains reverse transcriptase, RNase H, integrase and HIV protease[34,36]. mRNAs containing the slippery sequence and HIV FSS undergo frameshifting in bacterial (*E. coli*) ribosomes in vitro and in vivo at frequencies comparable to those observed for HIV frameshifting in eukaryotic translation systems[38–41] providing evidence for a common mechanism of frameshifting and ribosomal pausing induced by FSS in bacteria and eukaryotes. Hence, the FSS from HIV can be studied in *E. coli* system.

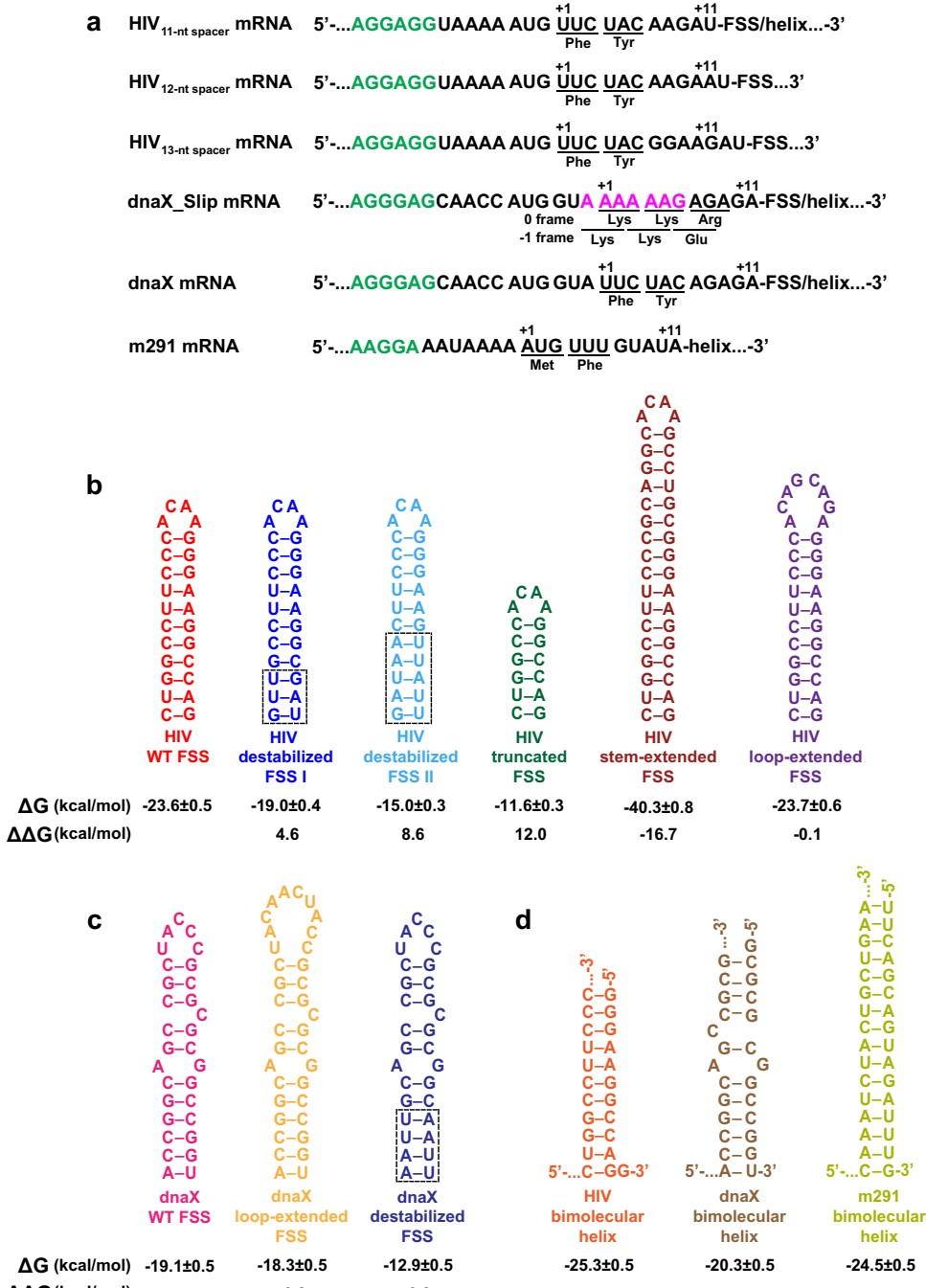

**Fig. 1 Model mRNAs containing either an FSS stem-loop or a bimolecular helix. a** Sequences of model mRNAs containing the Shine-Dalgarno (SD) sequences (green) and the FSS. The dnaX_Slip mRNA contains native slippery sequence (magenta). In other mRNA constructs, the slippery sequence is replaced with non-slippery codons. The first nucleotide of P-site codon and the last nucleotide bound in mRNA tunnel are labeled +1 and +11, respectively. **b-d** Secondary structures of frameshift-stimulating stem-loops (**b**, **c**) or bimolecular RNA helices (**d**). Dashed boxes indicate replacements of native GC basepairs with less stable A-U or G-U pairs. $\Delta G°$ values represent the overall thermodynamic stability of corresponding structures. $\Delta\Delta G°$ values of FSS variants indicate the changes in stabilities relative to the corresponding WT hairpins. Secondary structures and $\Delta G°/\Delta\Delta G°$ values (**b-d**) were determined by the RNAstructure software[45] using the default settings. Uncertainties in the folding free energy changes are determined using the variance and covariance of the nearest-neighbor parameter terms, determined by propagating the experiment uncertainties[62].

To probe properties of FSS stem-loops that are essential for inhibiting translocation and A-site tRNA binding, we introduced mutations into the FSS in model mRNAs that were derived from the *E. coli dnaX* and HIV *gag-pol* transcripts (Fig. 1, Supplementary Table 1)[31]. Frameshift changes the spacing between the P-site codon and the FSS and may also partially alleviate ribosome pausing. To delineate the effect of the FSS RNA hairpin on

ribosome pausing from frameshifting, slippery codons in sequences derived from HIV and dnaX mRNAs were replaced with "non-slippery" codons as previously described[31,42,43] (Fig. 1a). Mutations in the slippery sequence have been shown to decrease frameshifting efficiency to low or undetectable levels[26,29,32,37,44]. Each model mRNA contained the Shine–Dalgarno (SD, ribosome-binding site) sequence, a short

open reading frame (ORF) and a downstream FSS. In addition, upstream of the SD sequence, each mRNA contained a 25 nucleotide-long sequence complementary to a biotin-conjugated DNA oligonucleotide that is used to tether the mRNA to a microscope slide for smFRET experiments (Fig. 1a, Supplementary Table 1).

**The stability of three basepairs of the FSS first unwound by the ribosome has no effect on A-site tRNA binding**. We examined the FSS-mediated inhibition of the A site tRNA binding using a filter-binding assay as previously described[31]. Using this approach, we found that when positioned 11 or 12 nucleotides (but not 13 nucleotides) downstream of P-site codon, the FSS from HIV and dnaX strongly hinders binding of aminoacyl(aa)-tRNA to the A site[31]. Here, we measured kinetics of tRNA binding to the A site of ribosomes, which contained P-site $N$-Ac-Phe-tRNA$^{Phe}$ and were programmed with dnaX or HIV mRNA variants. In these ribosomal complexes, the FSS was positioned 11 nucleotides downstream from the P-site (UUC) codon (Fig. 1a). Throughout the manuscript, the position of the FSS is described relative to the first nucleotide of the P-site codon. For example, positioning 11 nucleotides downstream of the P-site codon means that the first nucleotide of the P-site codon and the first nucleotide of the FSS are +1 and +12, respectively (Fig. 1a).

The dnaX– or HIV–ribosome complexes were incubated with [³H]Tyr-tRNA$^{Tyr}$ and EF-Tu GTP (Fig. 2a). Next, the ribosomes were loaded onto a nitrocellulose filter and unbound [³H]Tyr-tRNA$^{Tyr}$ was washed away. Consistent with our previously published data[31], both wild-type dnaX and wild-type HIV FSSs severely slowed down the rate of [³H]Tyr-tRNA$^{Tyr}$ binding. The apparent pseudo-first-order rates of tRNA binding were 0.2 (HIV, Fig. 3a) and 0.3 (dnaX, Fig. 3b) min$^{-1}$, respectively. When ribosomes were programmed with truncated mRNAs lacking FSSs (either dnaX ΔFSS or HIV ΔFSS), the rate of [³H]Tyr-tRNA$^{Tyr}$ binding was too fast to be measured by the filter-binding assay requiring manual mixing of ribosome and tRNA.

Because the ribosome unwinds three pairs at a time, we investigated how the stability of the three bottom basepairs of the FSS affect tRNA binding. Two (in HIV) or three (in dnaX) G–C pairs at the bottom of respective FSSs were replaced with less stable G–U or A–U pairs to create HIV destabilized I and dnaX destabilized mRNAs (Fig. 1b, c). These mutations were predicted by the RNAstructure software[45] to decrease the stability of the bottom three basepairs in HIV and dnaX FSSs by 4.6 and 6.2 kcal/mol, respectively. However, probabilities of forming each basepair in HIV and dnaX stem-loops were predicted to be over 80% because of high overall stability of both HIV and dnaX mutant FSSs (−19.0 and −12.9 kcal/mol, respectively) and because sequences were designed to avoid pairing with the rest of the mRNA (Supplementary Fig. 1). To further destabilize the bottom of the HIV FSS, an additional two G–C basepairs were replaced with less stable A–U pairs to create HIV destabilized II mRNA (Fig. 1b, Supplementary Fig. 1). Relative to the wild-type FSS, the stability of the first five FSS basepairs in HIV destabilized II mRNA decreased by 8.6 kcal/mol while overall stability of the FSS remained high (−15.0 kcal/mol).

A-site tRNA binding was equally slow (0.2 min$^{-1}$) in the presence of the wild-type, destabilized I and destabilized II HIV FSSs (Fig. 3a). Likewise, experiments with dnaX mRNAs demonstrated that the destabilization of the bottom three basepairs has no effect on A-site tRNA binding (Fig. 3b). These results indicate that the stability of basepairs of the FSS that are first unwound by the ribosome does not affect the ability of HIV and dnaX FSS to inhibit A-site tRNA binding.

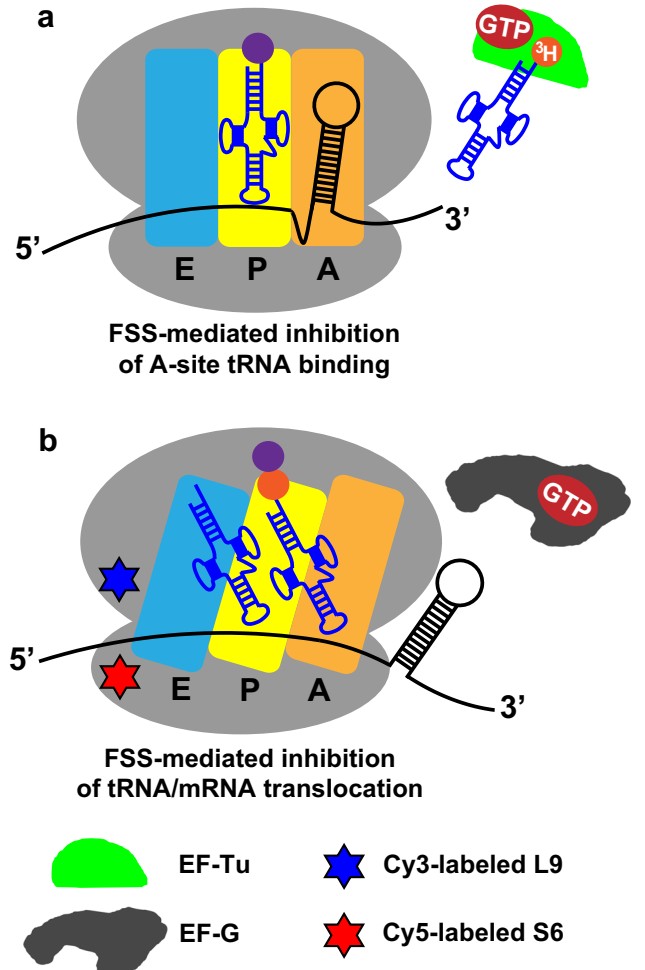

**Fig. 2 Experimental design. a** The effects of FSS variants on A-site tRNA binding were determined by filter-binding assays. 70S bacterial ribosome (gray) bound with P-site peptidyl-tRNA and mRNA was incubated with aminoacyl-tRNA, which had been charged with tritium-labeled amino acid and complexed with EF-Tu●GTP. **b** The effects of FSS variants on ribosome translocation coupled to intersubunit rotation were studied by smFRET. Cy5 (red hexagram) and cy3 (blue hexagram) fluorophores were attached to the 30S protein S6 and 50S protein L9, respectively. The pre-translocation S6-cy5/L9-cy3-labeled ribosome contained deacylated and peptidyl-tRNAs that adoped P/E and A/P hybrid states in the R conformation of the ribosome. EF-G●GTP was delivered to the ribosome by injection. The amino acids bonded with cognate tRNAs are shown in purple and orange. A, P and E sites of the ribosome are indicated by orange, yellow, and blue, respectively.

**The stem length and the presence of the loop are important for the ability of the FSS to inhibit A-site tRNA binding**. To examine the contribution of the stem-loop topology to FSS-induced inhibition of tRNA binding, we truncated the HIV FSS upstream of the loop and annealed a 12-mer RNA oligo to create a "loopless" RNA bimolecular helix, which is identical to the stem of the HIV stem-loop. Likewise, by annealing an RNA oligo to the dnaX mRNA, which was truncated upstream of the FSS loop, the dnaX bimolecular helix similar to the stem of wild-type dnaX FSS was created (Fig. 1d). The dnaX bimolecular helix is predicted to have two structures in equilibrium with folding free energy changes of −20.3 and −20.2 kcal/mol (Fig. 1d, Supplementary Fig. 1). The conformational difference is whether the C bulges in the mRNA (Fig. 1c, Supplementary Fig. 1) or the trans

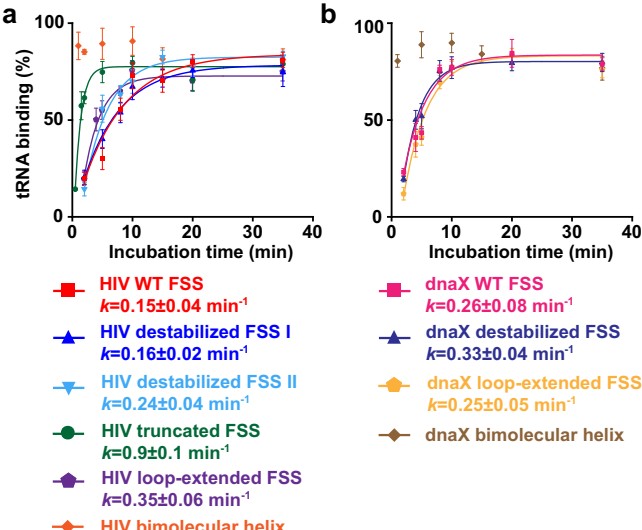

**Fig. 3 The stem length and the presence of the loop are more important for the FSS-induced inhibition of tRNA binding than FSS local and overall thermodynamic stabilities.** Kinetics of EF-Tu-catalyzed [³H]Tyr-tRNA^Tyr binding to the A site of the 70S ribosome. The ribosomes contained *N*-Ac-Phe-tRNA^Phe in the P site and were programmed with HIV (**a**) or dnaX (**b**) mRNAs, which contained an FSS variant (Fig. 1) 11 nucleotides downstream of the P-site codon as indicated. The binding of radio-labeled [³H]Tyr-tRNA^Tyr to ribosomes programmed with FSS-containing mRNA is shown relative to that observed in ribosomes programmed with corresponding ΔFSS mRNA. Apparent pseudo-first-order tRNA binding rates were deduced from single-exponential fitting as shown by the line graphs. Data are presented as mean values ± standard deviations of three independent measurements. Source data are provided as a Source Data file.

oligonucleotide (more similar to the dnaX hairpin structure; Fig. 1d, Supplementary Fig. 1). The base of the helix has similarly high probability as the dnaX wild-type hairpin stem-loop. When compared to the stem-loops, the bimolecular helices lack the loops.

Based on the free energy change for the formation of the bimolecular helices, we calculated the equilibrium constants and concentrations of the annealed bimolecular helices (Supplementary Table 2). These assessments predict that at concentrations of mRNAs and RNA oligos used during the ribosome complex assembly, nearly 100% of mRNA molecules form bimolecular helixes. Furthermore, estimated dissociation rates[46] (Supplementary Table 2) predict that both HIV and dnaX bimolecular helices remain stable within the duration of our biochemical and single-molecule experiments.

When ribosomes were programed with either HIV bimolecular helix or dnaX bimolecular helix mRNA, no inhibition of A-site tRNA binding was observed, i.e. the rate of tRNA binding was too fast to be measured by filter-binding assay, which involves manual mixing of the ribosome and tRNA (Fig. 3a, b). Hence, our results show that the presence of a loop in the FSS may be essential for the inhibition of A-site tRNA binding.

Loop regions of the wild-type HIV (ACAA) and dnaX (UACCC) FSSs differ in both sequence and length. Hence, despite the importance of the presence of the loop for the FSS-induced inhibition of A-site tRNA binding, specific loop sequence and length are unlikely to play key roles in the FSS-mediated regulation of translation elongation. To further test this hypothesis, we replaced the original, 4 nucleotide-long ACAA loop of HIV FSS with an 8 nucleotide-long ACAGCAGA loop, which was predicted to not lower the probability of the base pairs in the stem-loop (Supplementary Fig. 1). This loop extension resulted in a two-fold increase in the rate of A-site tRNA binding when compared to the wild-type HIV FSS (Fig. 3a, Supplementary Table 3). Replacing the 5 nucleotide-long UACCC loop of dnaX FSS with a 10 nucleotide-long UACAACUACC loop, which was predicted to not lower the probability of the base pairs in the stem-loop (Supplementary Fig. 1), did not affect the ability of the dnaX FSS to inhibit A-site binding (Fig. 3b, Supplementary Table 3). The relatively modest effect (HIV mRNA) or lack thereof (dnaX mRNA) of the loop extension on tRNA binding rate supports the idea that loop sequence and length are not critical for the ability of the FSS to inhibit A-site tRNA binding.

To explore the role of stem length, we truncated the wild-type HIV FSS by deleting the upper six basepairs adjacent to the loop while leaving bottom six basepairs intact (Fig. 1b). Importantly, the fractions of mRNA molecules forming all basepairs in the truncated stem-loop was predicted to be over 99% providing evidence that truncated HIV FSS variant is formed despite the decrease in overall thermodynamic stability from −23.6 kcal/mol (wild-type HIV) to −11.6 kcal/mol. Importantly, in the HIV mRNA with truncated FSS, the local stability of bottom basepairs first unwound by the ribosome was equal to the stability of these basepairs in the wild-type FSS. Because thermodynamic stability of RNA helixes is a function of helix length (number of basepairs)[47], it is impossible to truncate a stem-loop without lowering overall stem-loop stability. Nevertheless, the overall thermodynamic stability of the truncated HIV FSS (−11.6 kcal/mol) was only slightly lower that the stability of HIV destabilized FSS II (−15 kcal/mol, Fig. 1b) and dnaX destabilized FSS (−12.9 kcal/mol, Fig. 1c). Because the dnaX FSS contains two bulges (unpaired nucleotides in the stem) designing mutant versions of dnaX FSS with a truncated RNA stem while keeping the FSS sufficiently stable was not feasible. For that reason, we varied the stem length only in the HIV FSS.

When ribosomes were programmed by the HIV truncated FSS mRNA instead of the HIV wild-type FSS mRNA, the pseudo-first-order rate constant for tRNA binding increased from 0.2 to 0.9 min⁻¹ (Fig. 3a, Supplementary Table 3). Therefore, truncation of the HIV FSS substantially alleviates FSS-induced inhibition of A-site tRNA binding.

We also extended the upper stem of the wild-type HIV FSS by six basepairs. The probability of forming new basepairs in the stem-extended FSS was predicted to be over 99%, while the overall thermodynamic stability was computed to increase from −23.6 kcal/mol (wild-type HIV) to −40.3 kcal/mol (Fig. 1b). When we performed the filter-binding assay in the presence of the stem-extended HIV mRNA, we found that the extended FSS inhibits P-site tRNA binding. Non-enzymatic P-site binding of *N*-Ac-Phe-tRNA^Phe (which pairs with UUC codon 11 nt upstream of the FSS) and *N*-Ac-Met-tRNA^Met (which pairs with AUG codon 14 nt upstream of the FSS) were reduced to 50% and 70% of ribosome occupancy, respectively (Fig. 4, Supplementary Fig. 2a). It is possible that under our in vitro conditions, the stem-extended FSS binds directly to the P site in a fraction of ribosomes and hinders P-site tRNA binding.

Because ribosome complex assembly at the AUG codon was more efficient than assembly at the UUC codon of the extended FSS HIV mRNA, we examined the effect of the stem-extended FSS on A-site binding by measuring incorporation of radio-labeled aa-tRNAs during translation through three (Met, Phe, and Tyr) codons. We found that Tyr-tRNA^Tyr binding was equally efficient relative to incorporation of Met and Phe into polypeptide chain (Fig. 4). Likewise, kinetics of Tyr-tRNA^Tyr A-site binding to ribosomes containing P-site *N*-Ac-Met-Phe-tRNA^Phe was not affected as the rate of tRNA binding was too fast to be measured by a filter-binding assay, which involves manual mixing of the

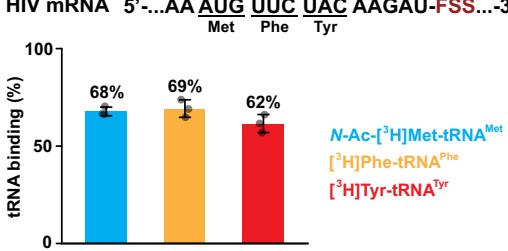

**Fig. 4 Extension of the stem length eliminates the FSS-induced inhibition of tRNA binding.** The ribosomes, which were programmed with HIV stem-extended FSS mRNA and contained P-site N-Ac-Met-tRNA$^{Met}$, were incubated with EF-G GTP, EF-Tu GTP, Phe-tRNA$^{Phe}$, and Tyr-tRNA$^{Tyr}$ for 5 min before loading ribosomes onto a nitrocellulose filter and washing away unbound aa-tRNA. The experiment was repeated three times with one of the three aminoacyl-tRNAs radio labeled, i.e. using N-Ac-[$^3$H]Met-tRNA$^{Met}$ (blue bar), [$^3$H]Phe-tRNA$^{Phe}$ (orange bar), or [$^3$H]Tyr-tRNA$^{Tyr}$ (red bar). Similar experiments were performed with ribosomes programmed with HIV ΔFSS mRNA. The binding of each radio-labeled tRNA to ribosomes programmed with the HIV stem-extended FSS mRNA is shown relative to that observed in ribosomes programmed with the HIV ΔFSS mRNA. Data are presented as mean values ± standard deviations of three independent measurements. Source data are provided as a Source Data file.

ribosome and tRNA (Supplementary Fig. 2b). Therefore, in contrast to the wild-type HIV FSS, the stem-extended HIV FSS does not hamper A-site tRNA binding when the FSS is positioned 11 nucleotides downstream of the P-site codon.

Taken together, our filter-binding data indicate that the specific stem length and the presence of the loop are important for the ability of the frameshift-inducing stem-loops to inhibit A-site tRNA binding while overall and local thermodynamic stability have negligible effects.

**The overall and local thermodynamic stabilities of the FSSs moderately affect ribosomal translocation.** When positioned 11 or 12 nucleotides downstream of the P-site codon, HIV and dnaX FSSs strongly inhibit A-site binding but do not completely block it as the A site eventually becomes filled during extended incubation of the ribsome with aminoacyl-tRNA and EF-Tu GTP (Fig. 3). The resulting pre-translocation ribosomes can be employed to examine the effect of the FSSs on the translocation step of the elongation cycle.

We next used single-molecule miscroscopy (Fig. 2b) and all aformenetioned mRNA constructs (Fig. 1) to investigate the properties of HIV and dnaX FSSs, which are important for the inhibition of ribosomal translocation. Following aminoacyl-tRNA binding to the ribosomal A site and peptide-bond formation, the pre-translocation ribosome predominantly adopts the rotated (R) conformation[48–50]. In this conformation, the small ribosomal subunit (the 30S subunit in bacteria) is rotated by ~9° relative to the large subunit (the 50S subunit)[51–53], and two tRNAs adopt the intermediate hybrid states of binding[54–56]. EF-G-catalyzed mRNA/tRNA translocation on the small subunit is coupled to the reverse rotation of the ribosomal subunits, restoring the nonrotated (NR) conformation in the post-translocation ribosome[48,50,57]. Hence, ribosome translocation is accompanied by R to NR transition. To follow the R to NR transition, which is coupled to translocation, we measured single-molecule Förster resonance energy transfer (smFRET) between fluorophores attached to the 50S protein bL9 and the 30S protein bS6[31] (Fig. 2b). The NR and R conformations of the ribosome have

been shown to correspond to the 0.6 and 0.4 FRET states of S6-cy5/L9-cy3 FRET pair[48,49].

The pretranslocation S6-cy5/L9-cy3-labeled ribosomes containing A-site peptidyl-tRNA (N-Ac-Phe-Tyr-tRNA$^{Tyr}$) and P-site deacylated tRNA$^{Phe}$ were programed with a FSS-containing mRNA (Fig. 2b). In these complexes, the FSS was positioned 11 nt downstream of the P-site codon. When the first nucleotide of the P-site codon is defined as +1, the +11 nucleotide is the last mRNA residue bound in the mRNA tunnel of the small subunit[1–3,58]. The first nucleotide of the FSS (+12) is positioned at the tunnel entrance (Fig. 1a).

Consistent with previous reports[49,59], when imaged in the absence of EF-G, pretranslocation ribosomes containing deacylated tRNA$^{Phe}$ in the P site spontaniously fluctuate between R and NR conformations at rates of 0.8 s$^{-1}$ (NR to R, Supplementary Fig. 3a) and 0.5 s$^{-1}$ (R to NR, Supplementary Fig. 3b). The R confromation is predominant as 95% of pretranslocation ribosomes spend <4 s in the NR confromation (Supplementary Fig. 3a). In contrast to these unproductive fluctuations, EF-G-induced translocation fixes the ribosome in the NR state[49]. To distinguish translocation from spontaneous unproductive fluctuations between R and NR conformations, we define EF-G-induced translocation as a transition from R to the stable NR state (i.e. 0.6 FRET state lasting over 4 s).

In translocation measurements, at 10 s of imaging, EF-G•GTP was injected into the microscope slide bound with pretranslocation ribosomes (Figs. 2b, 5). The dwell time $\tau_{trl}$(translocation) between the injection and the transition from R (0.4 FRET) to the stable NR (0.6 FRET) ribosomal conformations reflects the translocation rate (Fig. 5)[31].

When the HIV FSS was positioned 11 nucleotides downstream of the P site codon, translocation was so slow that most ribosomes remained in the R state until photobleaching of the acceptor dye. Under our experimental conditions, the rate of photobleaching was 0.03 s$^{-1}$ (Supplementary Fig. 3c). To extend the lifetime of the acceptor fluorophore and detect translocation in most ribosomes, the excitation laser was switched off after the EF-G injection and switched back on either 40 or 60 s later (Fig. 5a). Varying the interval for which the excitation laser was switched off between 40 and 60 s did not significantly affect the distribution of $\tau_{trl}$ in ribosome population indicating that the number of ribosomes translocating during the laser shutoff was negligible (Supplementary Fig. 4a, b). $\tau_{trl}$ measured in ribosomes programmed with the HIV FSS was remarkably long (Figs. 5a, 6a, Supplementary Fig. 5a) with median value of 73.3 s (Figs. 5b, 6a, Supplementary Fig. 4a, b). By contrast, when ribosomes were programmed with HIV mRNA lacking the FSS (HIV ΔFSS mRNA), median $\tau_{trl}$ was 2.8 s indicating that HIV FSS reduces the translocation rate by 25 folds (Figs. 5c, d, 6a, Supplementary Fig. 5a).

Next, we measured translocation kinetics in ribosomes programmed with dnaX mRNA, in which the dnaX FSS was positioned 11 nt downstream of the P-site (UUC) codon (Fig. 1a). Because of the slow rate of translocation in the presence of the dnaX FSS, the excitation laser was switched off after the EF-G injection and switched back on either 40 or 60 s later to extend lifetime of the acceptor fluorophore. Similar to experiments with HIV mRNA, varying the laser shutoff time did not significantly affect the distribution of $\tau_{trl}$ in ribosome population indicating that the number of ribosomes translocating during the laser shutoff was negligible (Supplementary Fig. 4e, f). When compared to the mRNA lacking the FSS, dnaX FSS reduced the translocation rate 15 times as the median of $\tau_{trl}$ changed from 5.1 (dnaX ΔFSS mRNA, Fig. 6b, Supplementary Fig. 6a) to 73.4 s (dnaX wild-type FSS mRNA, Fig. 6b, Supplementary Fig. 6b).

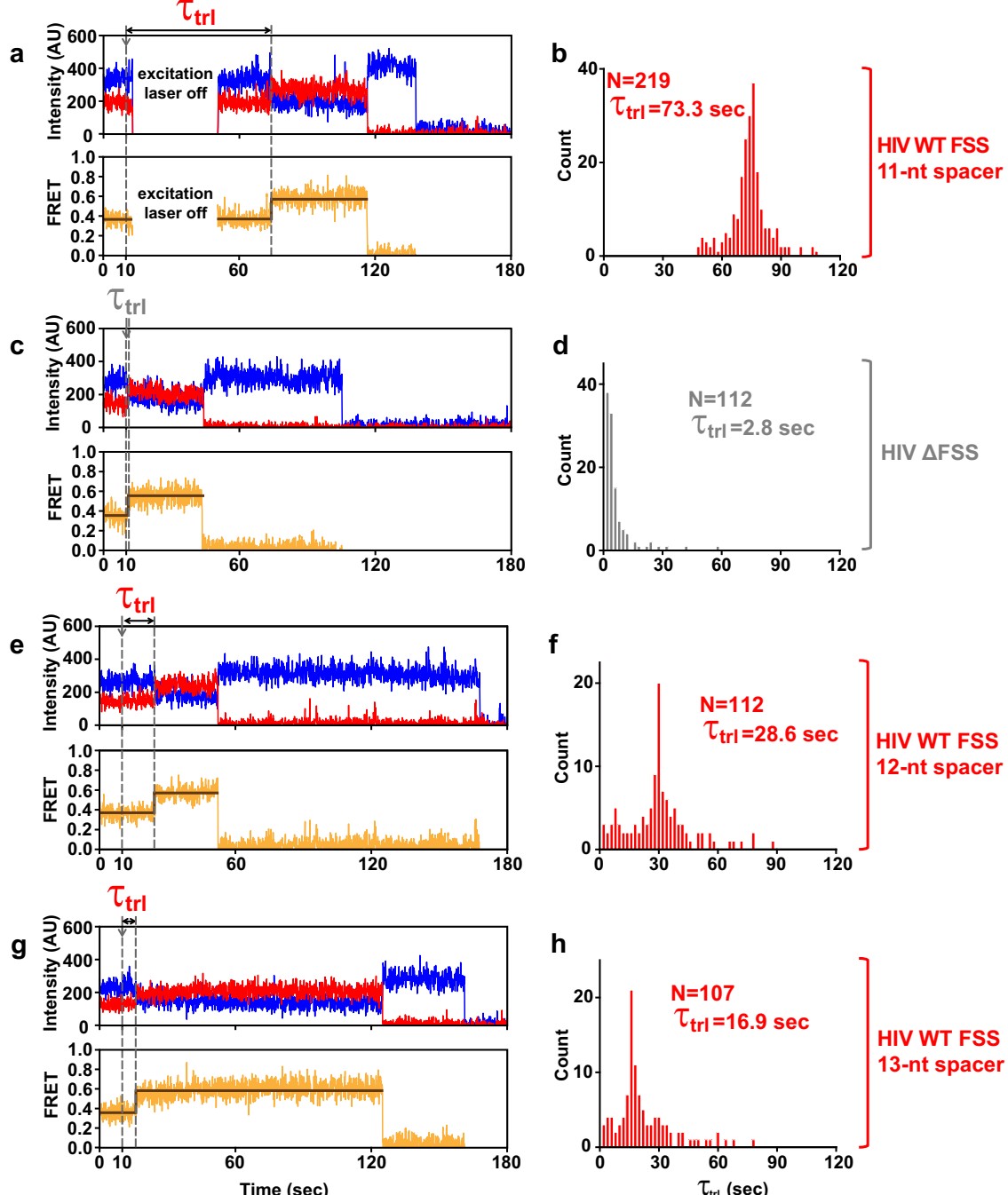

**Fig. 5 The FSS-induced inhibition of translocation is influenced by length of the mRNA spacer between P-site codon and FSS.** The pre-translocation S6-cy5/L9-cy3-labeled ribosomes were programmed with either HIV WT FSS (**a**, **b**, **e**, **f**, **g**, **h**) or HIV ΔFSS (**c**, **d**) mRNA (Fig. 1). The HIV FSS was positioned either 11 (**a**, **b**), 12 (**e**, **f**) or 13 (**g**, **h**) nucleotides downstream of the P-site codon, respectively. **a**, **c**, **e**, **g** Representative FRET traces show cy3 donor fluorescence (blue), cy5 acceptor fluorescence (red), and FRET efficiency (orange) fitted by two-state HHM (brown). Arrows show the injection of EF-G●GTP. The EF-G-catalyzed translocation corresponds to the transition from R (0.4 FRET) to a stable (i.e. lasting over 4 s) NR (0.6 FRET) state of the ribosome. **b**, **d**, **f**, **h** Histograms (2 s binning size) show the distributions and median values of the dwell time between the injection and the translocation, $\tau_{trl}$. $N$ indicates the number of traces assembled into each histogram. To extend lifetime of the acceptor fluorophore in the experiments with 11 nucleotide spacer mRNA, the excitation laser was turned off after the EF-G injection and switched back on either 40 s (**a**) or 60 s later. $\tau_{trl}$ distribution in (**b**) was combined from data sets acquired with 40 s (Supplementary Fig. 4a) and 60 s (Supplementary Fig. 4b) laser shut-off times. Source data are provided as a Source Data file.

To examine how the kinetics of translocation depend on the spacing between the P-site codon and the FSS, we made two HIV mRNA variants where the spacer between the P-site (UUC) codon was extended from 11 to 12 or 13 nucleotides (Fig. 1a, 5). Extending the P-site-FSS spacer from 11 to 12 nucleotides

decreased median $\tau_{trl}$ from 73.3 to 28.6 s (Fig. 5e, f) while lengthening the the P-site-FSS spacer to 13 nucleotides further decreased median $\tau_{trl}$ to 16.9 s (Fig. 5g,h). Hence, extending the P-site-FSS spacer progressively accelerates translocations. Noteworthy, in the experiments with 12 and 13-nucleotide long

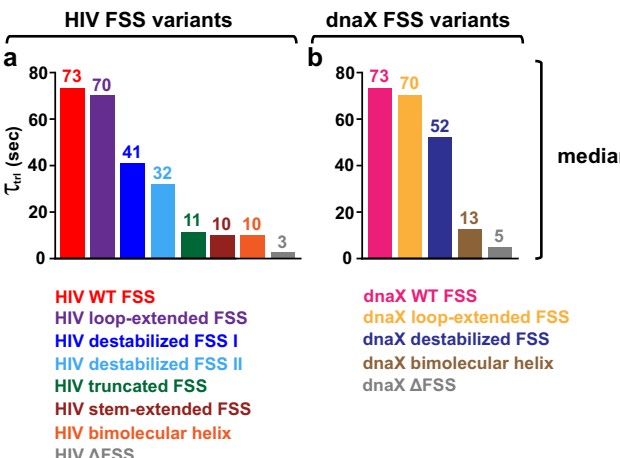

**Fig. 6 The stem length and the presence of the loop are more important for the FSS-induced inhibition of translocation than FSS local and overall thermodynamic stabilities.** The S6-cy5/L9-cy3 ribosomes were programmed with mRNA containing either an HIV (**a**) or dnaX (**b**) FSS variant as indicated (Fig. 1). Bar graphs show median values of $\tau_{trl}$. Respective mean values of $\tau_{trl}$ together with standard error of the mean (SEM) and standard deviations (SD) are shown in Supplementary Fig. 5. For all FSS variants but the HIV stem-extended FSS, $\tau_{trl}$ represents the dwell time between injection of EF-G•GTP and ribosome translocation in smFRET experiments. For the HIV stem-extended FSS, $\tau_{trl}$ corresponds to the dwell time between tRNA binding and translocation (Fig. 7). Source data are provided as a Source Data file.

P-site-FSS spacers, the excitation laser was not switched off at any point (Fig. 5e, g) because translocation rates were much faster than the acceptor photobleaching rate.

Assuming that all FSS nucleotides residing outside of the mRNA tunnel are basepaired, extending the P-site-FSS spacer from 11 to 12 or 13 nucleotides is expected to reduce the number of basepairs unwound upon a round of ribosome translocation from 3 to 2 or 1, respectively. Hence, our data may indicate that the number of basepairs unwound by the ribosome determines the extent of translocation pausing induced by the FSS. Alternatively, the short (11 nucleotide) spacer restricts the FSS dynamics and thus decreases the entropic penalty for forming FSS–ribosome interactions that inhibit translocation.

To investigate how the stability of three lowest basepairs of the FSS that are first unwound by the ribosome affects FSS-mediated inhibition of translocation, we used mutant variants of HIV and dnaX mRNAs (Fig. 1b, c). These mRNA variants were also employed in filter-binding experiments described above (Figs. 3 and 4). Destabilization of the three (HIV FSS) or four (dnaX FSS) bottom basepairs of the FSS that are first unwound by the ribosome slightly accelerated translocation as $\tau_{trl}$ decreased from 73 s (both wild-type HIV and wild-type dnaX FSSs) to 41.1 s (HIV destabilized FSS I, Fig. 6a, Supplementary Figs. 5a, 7a) and 52.3 s (dnaX destabilized FSS, Fig. 6b, Supplementary Figs. 5b, 6c), respectively. Reducing the stability of the first five FSS basepairs in HIV destabilized II mRNA led to two-fold acceleration of translocation rate as median $\tau_{trl}$ decreased from 73.3 s (wild-type HIV FSS) to 32.1 s (HIV destabilized FSS II) (Fig. 6a, Supplementary Figs. 5a, 7b).

When compared to the wild-type FSS HIV mRNA, distributions of $\tau_{trl}$ times observed in the presence of destabilized FSS I and II HIV mRNAs were markedly more heterogeneous (Supplementary Fig. 7a, b). These data likely indicate that destabilizing mutations introduced into HIV FSS and interactions of mutant FSS variants with the ribosome lead to structural

heterogeneity of the stem-loop such as possible fraying of the stem-loop bottom basepairs. Nevertheless, the predominant peaks in $\tau_{trl}$ distributions observed with destabilized HIV FSS I and II mRNAs (Supplementary Fig. 7a, b) likely correspond to the slowest ribosomes bound with the fully intact FSS (Fig. 5b).

Taken together, experiments with dnaX and HIV mRNA variants with destabilized bottom basepairs provide evidence that the stability of basepairs, which are first unwound by the translating ribosome, moderately contributes to the inhibition of translocation by the dnaX and HIV FSSs.

**The stem length and the presence of the loop are important for the inhibition of translocation by the FSSs.** We next explored how stem length and the presence of the loop affect FSS-induced inhibition of ribosome translocation. In contrast to moderate effects of destabilization of bottom basepairs, truncation of the HIV FSS by deleting the upper six basepairs adjacent to the loop while leaving the "bottom" six basepairs intact (Fig. 1b), accelerated translocation rate by 7 fold as $\tau_{trl}$ decreased from 73 to 11 s (Fig. 6a, Supplementary Fig. 7c).

We also examined how encountering the HIV FSS extended by six basepairs affects ribosome translocation. To that end, S6-cy5/ L9-cy3 labeled ribosomes bound with P-site N-Ac-Met-Phe-tRNA$^{Phe}$ and programmed with the stem-extended FSS HIV mRNA were imaged for 10 s. Then, EF-Tu•GTP•Tyr-tRNA$^{Tyr}$ and EF-G•GTP were injected to bind Tyr-tRNA$^{Tyr}$ to the A-site Tyr (UAC) codon and induce tRNA/mRNA translocation, respectively. Before the injection, ribosomes containing P-site peptidyl-tRNA (N-Ac-Met-Phe-tRNA$^{Phe}$) were predominantly observed in the NR (0.6 FRET) state. After the injection, the ribosomes showed an NR (0.6 FRET)-to-R (0.4 FRET) transition resulting from tRNA binding followed by the transpeptidation reaction and subsequent movement of tRNAs into hybrid states. The dwell time between the injection and the transition from the NR (0.6 FRET) to R (0.4 FRET) state, $\tau_{bd}$, primarily reflects the rate of Tyr-tRNA$^{Tyr}$ binding to the A site of the ribosome[31]. The subsequent reverse transition from the R to the stable NR state (0.6 FRET lasting over 4 s) indicts translocation of mRNA and tRNA. The dwell time, $\tau_{trl}$, between the NR to R and R to NR transitions in our injection experiments reflects to the translocation rate (Fig. 7a–d).

Consistent with filter-binding experiments (Fig. 4), Tyr-tRNA$^{Tyr}$ binding time, $\tau_{bd}$, observed in the presence (4.3 s; Fig. 7b) and the absence of the stem-extended HIV FSS (3.1 s; Fig. 7e) were similar. These results indicate that the stem-extended HIV FSS positioned 11 nucleotides downstream of the P-site codon cannot dock into the A site and does not inhibit tRNA binding. Median translocation time, $\tau_{trl}$, observed in the presence and the absence of the stem-extended HIV FSS were 10.2 and 3.2 s, respectively (Fig. 7d, f). Hence, in contrast to the wild-type HIV FSS, which induces a 25-fold reduction in translocation rate, the stem-extended HIV FSS inhibited translocation by only 3 folds. These data demonstrate that the optimal stem length contributes more to the FSS ability to inhibit translocation than overall and local thermodynamic stabilities of the stem-loop.

Swapping the four nucleotide-long ACAA loop of the HIV FSS with an eight nucleotide-long ACAGCAGA loop (Fig. 1b) did not change median $\tau_{trl}$ (Fig. 6a, Supplementary Figs. 4c, d, 5a, 7d). Likewise, replacing the UACCC loop of the dnaX FSS with an extended UACAACUACC loop (Fig. 1c) did not change translocation rate (Fig. 6b, Supplementary Figs. 5b, 6d). These observations indicate that altering length and sequence of the FSS loop does not affect FSS-induced inhibition of translocation.

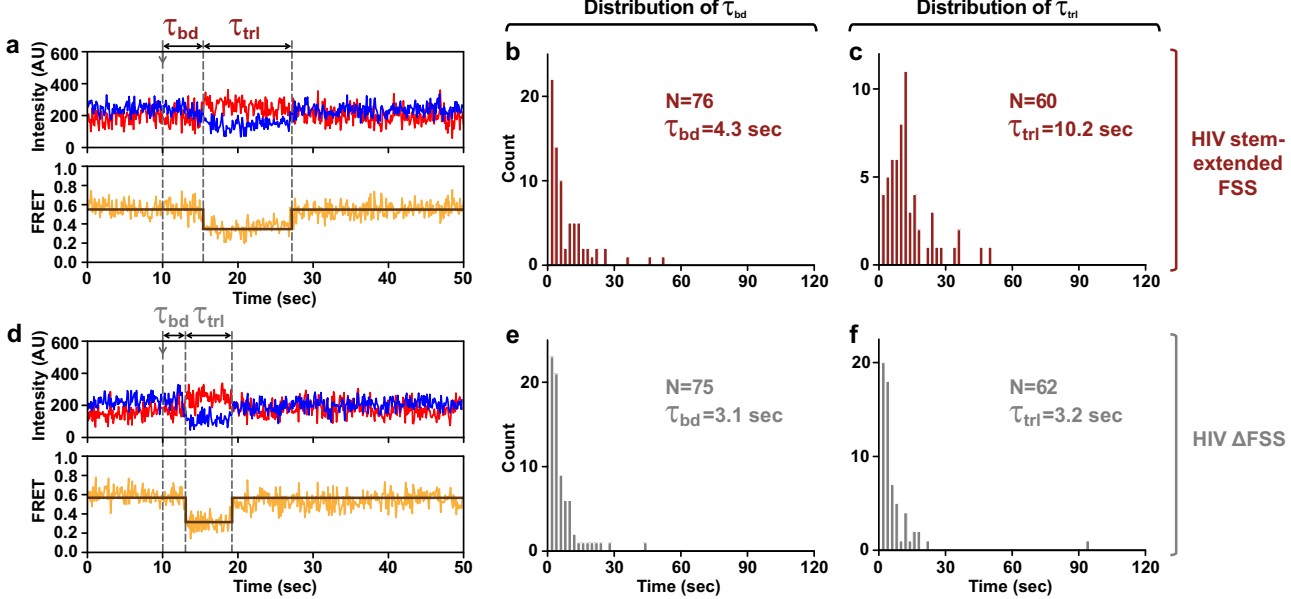

**Fig. 7 The FSS-induced inhibition of translocation is alleviated by extension of the stem length.** The S6-cy5/L9-cy3-labeled ribosomes containing P-site N-Ac-Met-Phe-tRNA$^{Phe}$ were programmed with either HIV stem-extended FSS (**a–c**) or HIV ΔFSS (**d–f**) mRNA (Fig. 1). The FSS was positioned 11 nucleotides downstream of the P-site codon. **a**, **d** Representative FRET traces show cy3 donor fluorescence (blue), cy5 acceptor fluorescence (red) and FRET efficiency (orange) fitted by two-state HHM (brown). Arrows mark the injection of EF-Tu•GTP•Tyr-tRNA$^{Tyr}$ and EF-G•GTP. $\tau_{bd}$ is the dwell time between the injection and the Tyr-tRNA$^{Tyr}$ binding to the ribosomal A site, which corresponds to the transition from NR (0.6 FRET) to R (0.4 FRET) state of the ribosome. $\tau_{trl}$ is the dwell time between the A-site tRNA binding and the EF-G-catalyzed ribosome translocation, which corresponds to the transition from R to the stable (i.e. lasting over 4 s) NR state of the ribosome. **b**, **c**, **e**, **f** Histograms (2 s binning size) show the distributions and median values of $\tau_{bd}$ and $\tau_{trl}$. N indicates the number of traces assembled into each histogram. The full-length views of the FRET traces are shown in Supplementary Fig. 8. Source data are provided as a Source Data file.

By contrast, replacing the wild-type HIV and dnaX FSSs with "loopless" bimolecular helices that are identical to stems of HIV and dnaX FSSs (Fig. 1d) similarly accelerated the rate of translocation by ~7 folds (Fig. 6a, b, Supplementary Fig. 5a,b). Median $\tau_{trl}$ decreased from 73 s (both wild-type HIV and dnaX FSSs) to 10.3 s (HIV bimolecular helix, Fig. 6a, Supplementary Fig. 7e) and 12.7 s (dnaX bimolecular helix, Fig. 6b, Supplementary Fig. 6e). These results show that the presence of the loop plays an important role in the FSS-induced inhibition of translocation.

Next, we tested whether another bimolecular RNA duplex that is not derived from a frameshift-inducing stem-loop has a similar effect on translocation as bimolecular helices derived from HIV and dnaX FSSs. To that end, a 16 nucleotide-long RNA oligo was annealed to an unstructured region of a model mRNA derived from the phage T4 gene *32* mRNA named m291[60] (Fig. 1a–d). Importantly, the overall thermodynamic stability of the m291 bimolecular helix (−24.5 kcal/mol) was comparable to the stabilities of bimolecular helixes derived from HIV and dnaX FSSs and was also slightly higher than the stabilities of wild-type HIV and dnaX FSSs (Fig. 1d).

Similar to experiments with HIV and dnaX bimolecular helices, in pretranslocation ribosomes programed with m291 mRNA containing A-site peptidyl-tRNA (N-Ac-Met-Phe-tRNA$^{Phe}$) and P-site deacylated tRNA$^{Met}$, the RNA oligo was positioned 11 nucleotides downstream of the P-site (AUG) codon (Fig. 1a). When compared to translocation observed in m291 mRNA-programmed ribosomes in the absence of the RNA oligo (Fig. 8a), the presence of bimolecular helix reduced the rate of translocation by 2 folds (Fig. 8b). Although HIV, dnaX and m291 bimolecular helixes differed in sequence and stabilities of the three basepairs, which are adjacent to 30S mRNA tunnel and unwound by the ribosome upon

translocation, median values of dwell time $\tau_{trl}$ measured in the presence of these constructs were similar.

Remarkably, when compared to the mRNAs lacking the FSS (m291, HIV ΔFSS and dnaX ΔFSS mRNAs), the truncated FSS, the extended FSS and the "loopless" bimolecular helix reduced translocation rate by ~2–3 folds. These results are consistent with previous optical tweezers experiments demonstrating that translocation through three GC basepairs is only 2–3-fold slower than translocation along a single-stranded codon[10,12,13].

Taken together, our data show that unwinding of three basepairs during one round of translocation only moderately slows down the ribosome as evident from experiments with bimolecular helices. Hence, much stronger inhibition of translocation observed in the presence of wild-type HIV and dnaX FSSs are due to their specific structural properies rather than their high thermodynamic stabilities.

**Effects of structural purturbations in FSS on frameshifting efficiency**. To examine how variations in the FSS structure affect ribosome frameshifting, we determined the efficiency of −1 PRF on the dnaX mRNA containing native slippery sequence (dnaX_Slip, Figs. 1 and 9) using the filter-binding assay, as we previously described[31]. The beginning of the ORF in the dnaX_Slip mRNA encodes Met-Val-Lys-Lys-Arg in the 0 frame and Met-Val-Lys-Lys-Glu in the −1 frame. Ribosomes bound with dnaX_Slip mRNA and P-site N-Ac-Val-tRNA$^{Val}$ were incubated with EF-G•GTP, EF-Tu•GTP, Lys-tRNA$^{Lys}$ (decodes the slippery sequence) and [$^3$H]Glu-tRNA$^{Glu}$ (binds in the −1 frame). Consistent with previous publications[29,31,32,37], we observed a frameshifting efficiency (corresponding to the [$^3$H] Glu-tRNA$^{Glu}$ occupancy of the ribosomal A site) of ~61% (Fig. 9). When ribosomes were programmed with the truncated dnaX_Slip ΔFSS mRNA, which lacks the FSS, the efficiency of −1

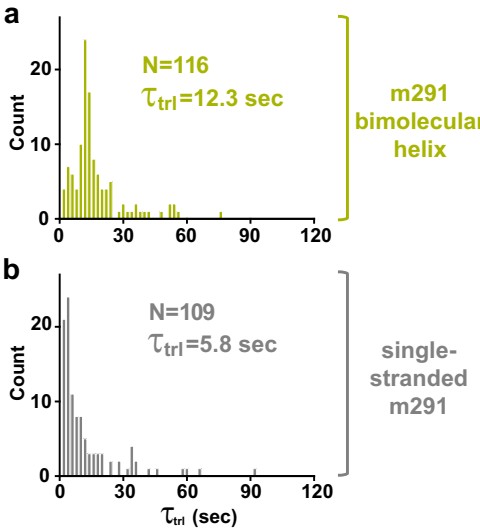

**Fig. 8 The bimolecular helix formed in the m291 mRNA at the entry to mRNA tunnel mildly inhibits translocation.** Kinetics of translocation was measured by smFRET experiments with pre-translocation S6-cy5/L9-cy3 ribosomes programmed with m291 mRNA containing a 16-bp bimolecular RNA helix (Fig. 1) positioned 11 nucleotides downstream of P-site codon (**a**) or with single-stranded m291 mRNA (**b**). Histograms (2 s binning size) show the distributions and median values of $\tau_{trl}$, which represents the dwell time between injection of EF-G GTP and ribosome translocation. $N$ indicates the number of FRET traces compiled into each histogram. Source data are provided as a Source Data file.

PRF decreased to ~18% (Fig. 9), demonstrating that the FSS stimulates ribosome frameshifting.

We next introduced into dnaX_Slip mRNA mutations that destabilize bottom basepairs of dnaX FSS and are identical to nucleotide substitutions, which we used in the context of non-slippery codons (Fig. 1). These mutations, which did not affect A-site tRNA binding and moderately accelerated ribosome translocation in the context of non-slippery codons, reduced frameshifting efficiency from 61% to 41% (Fig. 9). Replacing the dnaX FSS with a bimolecular helix, which is identical in sequence and stability to stem of wild-type dnaX FSS, decreased frameshifting efficiency from 61% to 25%. Stronger decline of frameshifting efficiency in the presence of the bimolecular helix is consistent with the experiments performed in the context of the non-slippery sequence, showing that replacing dnaX FSS with bimolecular helix dramatically diminishes FSS-induced inhibition of tRNA binding and translocation (Fig. 9). These results also suggest that frameshifting efficiency, at least to some degree, positively correlates with the length of FSS-induced translation pause.

## Discussion

We find that the total thermodynamic stablity of the FSSs and the local stablity of the basepairs adjacent to the entry of mRNA tunnel moderately influence ribosome pausing. In our work, we used nearest-neighbor estimation of helix stabilities, which is well established as accurate[47]. Using recent quantification of the confidence of nearest-neighbor parameters by propagating the experimental errors through the derivation of parameters[61,62], we found that the error estimates are smaller than the calculated $\Delta\Delta G$°s (Fig. 1). Hence, instead of merely acting as a physical barrier for the translating ribosome, the FSS likely hinder protein synthesis by forming specific interactions with the ribosome. Consistent with this idea, the lack of correlation between total

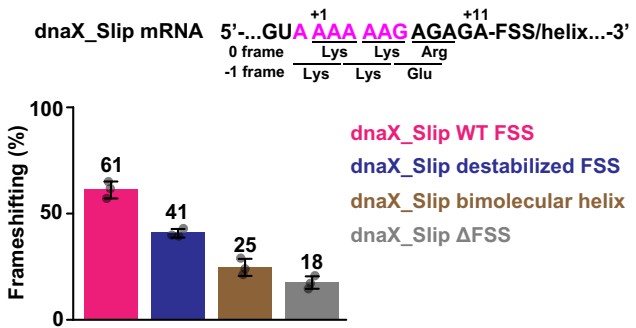

**Fig. 9 The presence of the loop is more important for the FSS-stimulated −1 PRF than the local and overall thermodynamic stabilities of the FSS.** Ribosomes containing P-site N-Ac-Val-tRNA$^{Val}$ were programmed with dnaX_Slip mRNA containing an FSS variant (Fig. 1). The ribosomes were incubated with EF-G•GTP, EF-Tu•GTP, Lys-tRNA$^{Lys}$ and [$^3$H]Glu-tRNA$^{Glu}$ (decodes GAG codon in −1 frame) for 5 min. For each FSS variant, [$^3$H]Glu-tRNA$^{Glu}$ binding to the ribosomal A site was measured by filter-binding assays. Frameshifting efficiency is represented by A-site occupancy by [$^3$H] Glu-tRNA$^{Glu}$ normalized by the P-site occupancy of N-Ac-[$^3$H]Glu-tRNA$^{Glu}$, which non-enzymatically binds to the ribosome programmed with dnaX_Slip ΔFSS mRNA. Data are presented as mean values ± standard deviations of three independent measurements. Source data are provided as a Source Data file.

thermodynamic stability of the FSS and the efficiency of frameshifting has been observed in several studies[63,64], including our own experiments (Fig. 9). In addition, significant destabilization of the bottom three basepairs of the HIV FSS has been shown to have no effect on the efficiency of frameshifting[34].

By contrast, stabilization of the bottom basepairs of the HIV FSS has been shown to substantially increase frameshifting efficiency[34]. Furthermore, positive correlation between frameshifting efficiency and the overall stability of dnaX FSS have been demonstrated by Atkins and colleagues[32]. Several factors, however, complicate the comparison of ribosome pausing observed in the context of non-slippery codons in our work with previously reported measurements of frameshifting. First, during translocation coupled with −1 PRF, two instead of three bottom basepairs of the FSS are unwound. Second, in contrast to short model mRNAs used in our experiments, FSS sequences in long mRNA encoding frameshifting reporters may form several alternative structures by forming basepairs with the rest of mRNA, which compete with FSS basepairs. It is possible that the mutations designed to destabilize the FSS resulted in alternative secondary structures, which complicate the analysis of the data. By contrast, stabilizing mutations in the FSS might increase frameshifting efficiency by enlarging the fraction of mRNA, which contains properly folded FSS, rather than by hindering FSS unwinding by the ribosome. Although the duration of FSS-induced ribosome pausing might weakly correlate with frameshifting efficiency, it is more likely that apparent discrepancies between measurements of ribosome pausing and frameshifting are due to formation of alternative secondary structures in long reporter mRNAs.

Our data show that structural properties of the FSS such as stem length and the presence of the loop make significant contribution to FSS-induced long ribosome pauses. Noteworthy, HIV and dnaX FSSs are 12 and 11 basepairs long, respectively (Fig. 1b, c). The idea that optimal stem length (around 11–12 basepairs) is important for ribosome pausing and frameshifting is supported by the study of the frameshift-inducing stem-loop derived from Simian retrovirus type-1 gag-pro transcript[65]. This work demonstrated that both increase and decrease in the length of the stem from the original length of 12 basepairs reduce frameshifting efficiency.

How do FSSs interact with the ribosome to inhibit translation elongation? Our recent studies indicated that the FSSs transiently inhibit tRNA biding by directly blocking the A site. In the structure of HIV FSS–ribosome complex, mRNA nucleotides between the P-site codon and the FSS are dissociated from the mRNA tunnel and the FSS is binds into the A site of the ribosome[31]. Extending HIV FSS by six basepairs likely prevents FSS docking into the A site because HIV stem-extended FSS does not inhibit tRNA binding (Figs. 4, 7, Supplementary Fig. 2b).

In the HIV FSS–ribosome structure, the FSS loop contacts the A-site finger (Helix 38) of 23S rRNA. Consistent with the importance of the FSS loop–ribosome interactions, HIV and dnaX bimolecular helixes lacking the loop do not inhibit A-site tRNA binding. Furthermore, shortening the stem-loop also likely eliminates the interaction of the loop with the A site finger, and thus partially alleviates the FSS-induced inhibition of A-site tRNA binding. In our experiments, variations in the sequence and length of the loop do not affect the ability of the HIV and dnaX FSS to inhibit tRNA binding (Fig. 3). Hence, the interactions between the FSS loop and the large subunit do not appear to be sequence specific and likely involve RNA phosphate backbone.

In addition to inhibiting A-site tRNA binding, the HIV and dnaX FSSs slow down translocation. This likely occurs when mRNA is threaded through the 30S mRNA channel and the FSS interacts with the entry to the mRNA tunnel that is lined by ribosomal protein uS3, uS4, and uS5. Such a conformation of the FSS was visualized by cryo-EM reconstructions of the dnaX FSS bound to bacterial ribosome[42] and frameshift-inducing pseudoknot from SARS-CoV-2 RNA genome bound to mammalian ribosome[66]. In the dnaX–ribosome complex, similar to the dnaX–ribosome complexes used in our work, the FSS was positioned 11 nucleotides downstream of the P-site codon[42]. Because of intermediate local resolution at the dnaX hairpin, molecular details of the FSS interactions with E. coli ribosome remain unclear. SHAPE probing of the dnaX FSS performed on this complex demonstrated that dnaX FSS is properly folded when adjacent to mRNA tunnel, although basepairing status of nucleotides +13, +14 and +15 of the FSS was ambiguous[42,58].

Our data support the assumption that when bound at the entry to the mRNA tunnel (i.e. 11 nucleotides downstream of the P-site codon in bacterial ribosome), the HIV and dnaX FSS remain fully folded. Mutations destabilizing basepairing interactions formed by nucleotides +12, +13 and +14 in the HIV FSS and +13, +14 and +15 in the dnaX FSS appreciably affected the rate of translocation in our smFRET experiments indicating that unwinding of these basepairs accompanies translocation (Fig. 6). $\tau_{trl}$ observed in the presence of HIV destabilized FSS I (41 s) and HIV destabilized FSS II (32 s) (Fig. 6a) were relatively similar, consistent with the idea basepairs formed by nucleotides +15 and +16 are not unwound during translocation in our smFRET experiments.

Similar to earlier proposals[66,67], we hypothesize that interactions of the FSS with the entry of mRNA tunnel inhibit ribosome translocation and helicase activity. Recent optical tweezers experiments indicated that unwinding of three basepairs adjacent to the mRNA tunnel occurs through two parallel pathways, one of which is 7-fold slower than the other[12]. An mRNA stem-loop likely samples alternative conformations at the entry to mRNA tunnel. In some of these conformations, translocation and base-pair unwinding might be slowed down. Unique interactions with the ribosome that distinguish the FSS from the bimolecular helix likely favor this slow translocation/unwinding pathway. Our data show that the presence of the loop and specific stem length stabilize the inhibitory conformation of the FSS-ribosome complex. The cryo-EM reconstruction of the SARS-CoV-2 FSS bound at the entry to mRNA tunnel of mammalian ribosome[66] indicated that the pseudoknot wedges between the head and the body domains of the small ribosomal subunit and thus might restrict the swiveling motion of the head domain relative to the body and platform domains of the small subunit that accompanies ribosome translocation[68]. Structures with higher local resolution near the FSS positioned at entry to the mRNA tunnel will be required to elucidate the molecular mechanism of FSS-mediated switching between the slow and fast pathways of translocation/mRNA unwinding observed in optical tweezers experiments.

Interestingly, when positioned 11 nucleotides downstream of the P-site codon, a bimolecular helix identical to the stem of the HIV FSS inhibits translocation less than the wild-type HIV FSS positioned 12 or 13 nucleotides downstream of the P-site codon. In other words, unwinding of three basepairs of the HIV bimolecular helix slows down the ribosome less than unwinding of two basepairs or one basepair of the HIV FSS. This observation is consistent with the interpretation that thermodynamic stabilities of basepairs adjacent to the entry to the mRNA tunnel do not account for the inhibitory effect of the FSS. Partial alleviation of the FSS-mediated pausing observed in our experiments when the spacing between the FSS and the P-site codon was extended from 11 to 12 or 13 nucleotides likely indicates that the short (11 nucleotide) spacer decreases entropic cost for forming the inhibitory interactions between the FSS and the ribosome by restricting the FSS dynamics.

Our findings have a number of implications for translational regulation at transcriptome-wide levels. On the one hand, because unwinding of three basepairs slows down the ribosome only 2–3 times, most elements of secondary structure are not expected to produce long ribosome pauses that are consistent with published ribosome profiling studies[14,15]. However, variations in the propensity to form basepairing interactions within the ORF may fine-tune the elongation rate and thus, at least to some extent, regulate the translational efficiency[16].

On the other hand, mRNA stem-loops whose specific stem length and loop structure enable them to either dock into the A site to block tRNA binding or to interact with the entry to mRNA channel to hinder translocation can produce extended ribosome pauses. These pauses might be more transient in live cells than the pauses observed in our in vitro experiments because combining the FSS with the non-slippery sequences may exacerbate the FSS-induced inhibition of translation[18,31]. Furthermore, cellular concentrations of EF-Tu and EF-G might be higher than those used in our experiments. Nevertheless, our experiments provide several lines of evidence that specific mRNA stem-loops inhibit translation much stronger than an mRNA helix in general.

Further investigation of structural properties of mRNA stem-loops and pseudoknots important for translation inhibition may inform identification of previously unknown regulatory secondary structure elements within mRNA ORFs that produce programmed ribosome pausing events. Our data also have important implications for the rational design of mRNA sequences for industrial applications, including mRNA vaccines. On the one hand, specific stem-loops and pseudoknots that induce extended ribosome pausing/stalling should be avoided in the design of mRNA vaccines because ribosome stalling triggers quality-control pathways, including ribosome rescue, nascent polypeptide degradation and no-go mRNA decay[69]. On the other hand, the PRF-stimulating mRNA secondary structures can be used to encode alternative polypeptide products in the same transcript via PRF, thus expanding the protein-coding capacity of the artificial mRNAs.

## Methods

**Preparation of ribosome, translation factors, tRNA and mRNA.** The tight couple 70S ribosomes were purified from E. coli MRE600 strain[48]. S6-cy5/L9-cy3 ribosomes were prepared by partial reconstitution of ΔS6-30S and ΔL9-50S

subunits with S6-41C-cy5 and L11-11C-cy3 proteins[48,70]. tRNAs (purchased from Chemical Block) were aminocylated using S100 enzyme according to the standard procedure[54]. Histidine-tagged EF-G and EF-Tu were expressed and purified by standard procedures. Sequences encoding dnaX and HIV mRNA variants (Supplementary Table 1) were cloned and prepared using T7 polymerase and run-off in vitro transcription[31]. The bimolecular RNA helices were prepared by annealing 3 μM mRNA with 6 μM complementary RNA oligo (synthesized by IDT, Supplementary Table 1) at 65 °C for 3 min in polyamine buffer (50 mM HEPES (pH 7.5), 6 mM $Mg^{2+}$, 6 mM β-mercaptoethanol, 150 mM $NH_4Cl$, 0.1 mM spermine, and 2 mM spermidine).

**smFRET measurements**. The total internal reflection fluorescence (TIRF) microscopy-based smFRET experiments were performed using published procedures[31,49,70] with modifications. Quartz slides were coated with dichlorodimethylsilane (DDS), bound with biotin-BSA and passivated by Tween-20[71]. 30 μL 0.2 mg/mL neutravidin dissolved in H50 buffer (20 mM HEPES (pH 7.5) and 50 mM KCl) was loaded and bound to the biotin-BSA. Ribosomal complexes were imaged in polyamine buffer (50 mM HEPES (pH 7.5), 6 mM $Mg^{2+}$, 6 mM β-mercaptoethanol, 150 mM $NH_4Cl$, 0.1 mM spermine, and 2 mM spermidine) with 0.8 mg/mL glucose oxidase, 0.625% glucose, 0.4 μg/mL catalase, and 1.5 mM 6-hydroxy-2,5,7,8-tetramethylchromane-2-carboxylic (Trolox). The ribosome complexes containing bimolecular RNA helices were imaged in the presence 1 μM complementary RNA oligo to assure oligo re-annealing. FRET data were acquired and processed using Single software and MatLab scripts (http://ha.med.jhmi.edu/resources/)[31,70]. The exposure time was 100 ms per frame. Rates of fluctuations between 0.4 and 0.6 FRET states were determined by idealizing FRET traces with a 2-state Hidden Markov model (HMM)[72]. Only those fluorescence vs time traces that showed single-step photobleaching for both Cy5 and Cy3 dyes and anti-correlated changes in Cy3 and Cy5 fluorescence were included in the analysis.

Ribosome complexes used in smFRET experiments were assembled as follows. 0.3 μM S6/L9-labeled ribosomes were incubated with 0.6 μM tRNA and 0.6 μM mRNA in polyamine buffer at 37 °C for 15 min. To fill the A site, 0.6 μM aminoacyl-tRNA were pre-incubated with 10 μM EF-Tu and 1 mM GTP in polyamine buffer at 37 °C for 10 min. To overcome the FSS-induced inhibition of tRNA binding, 0.3 μM ribosomal complex containing peptidyl-tRNA in the P site was incubated with 0.6 μM aminoacyl-tRNA (complexed with EF-Tu•GTP) at 37 °C for 10 min. The ribosome samples were loaded and immobilized on the slide as previously described[31]. To catalyze translocation, 1 μM EF-G•GTP (in imaging buffer) was delivered at 0.4 mL/min speed by syringe pump at 10 s of imaging. smFRET traces that showed a transition from 0.4 to stable (i.e. lasting over 4 s) 0.6 FRET state after EF-G injection were used to estimate the rate of translocation.

To prepare the ribosomes programmed with HIV stem-extended FSS mRNA, the N-Ac-Met-tRNA$^{Met}$ was bound to the P site of the S6/L9-labeled ribosome as described above. 0.3 μM N-Ac-Met-tRNA$^{Met}$-containing ribosome was incubated with 0.6 μM Phe-tRNA$^{Phe}$ (binding to the A site of the ribosome), 1 μM EF-G and 1 mM GTP at 37 °C for 10 min. After complex immobilization on the slide and removal of EF-G•GTP and unbound tRNAs[31], the ribosomes containing N-Ac-Met-Phe-tRNA$^{Phe}$ in the P site were imaged and a mixture of 1 μM of EF-Tu GTP Tyr-tRNA$^{Tyr}$ and 1 μM EF-G GTP (in imaging buffer) was delivered by pump injection at 10 s of imaging. Traces that showed consecutive transitions from 0.6 to 0.4 to stable (i.e. lasting over 4 s) 0.6 FRET state were used to estimate the rates of tRNA binding and translocation.

**Filter-binding assay**. Filter-binding assays were performed using standard procedures[31,73] with minor modifications. Ribosome complexes were assembled and incubated with aminoacyl-tRNA charged with radio-labeled amino acid as described previously[31]. Similar to the smFRET experiments described above, the complexes containing the bimolecular RNA helices were incubated in the presence of 1 μM complementary RNA oligo. The ribosomes programmed with HIV stem-extended FSS mRNA were, respectively, assembled with radiolabeled tRNAs (N-Ac-[$^3$H]Met-tRNA$^{Met}$, [$^3$H]Phe-tRNA$^{Phe}$, or [$^3$H]Tyr-tRNA$^{Tyr}$, as indicated in figure legends).

The ribosome complexes were applied to a nitrocellulose filter (MiliporeSigma), which was immediately washed with 500 μl (for dnaX–ribosome complexes) or 800 μl (for HIV– or m291–ribosome complexes) of ice-cold polyamine buffer containing 20 mM $Mg^{2+}$. The 20 mM $Mg^{2+}$ concentration was used to stabilize the ribosome complexes under non-equilibrium condition.

**Frameshifting assay**. To prepare the ribosome programmed with dnaX_Slip mRNA, 0.6 μM 70S ribosome was incubated with 1.2 μM N-Ac-Val-tRNA$^{Val}$ and 1.2 μM mRNA in polyamine buffer at 37 °C for 15 min. Subsequently, 0.6 μM dnaX_slip mRNA-programmed ribosomal complex containing peptidyl-tRNA in the P site was incubated with 2.4 μM Lys-tRNA$^{Lys}$, 1.2 μM [$^3$H]Glu-tRNA$^{Glu}$, 10.0 μM EF-Tu GTP, and 4.0 μM EF-G•GTP at 37 °C for 5 min. Binding of [$^3$H]Glu-tRNA$^{Glu}$ to the ribosome was quantified by filter-binding assay as described above. To calculate frameshifting efficiency, ribosome A-site occupancy by [$^3$H]Glu-tRNA$^{Glu}$ was normalized by the P-site occupancy of N-Ac-[$^3$H]Glu-tRNA$^{Glu}$

non-enzymatically bound to the ribosome programmed with dnaX_Slip ΔFSS mRNA.

**RNAstructure prediction**. RNAstructure (version 6.2) was used to predict the RNA secondary structure and thermodynamic stability of corresponding structures[45]. RNA secondary structures were predicted using maximum expected accuracy[74] from basepair probabilities estimated from a partition function calculation. In these calculations, nucleotides from the beginning of the Shine–Dalgarno sequence to nucleotide +11 downstream of the first nucleotide of the P-site codon and the oligonucleotide tether-binding site were forced to be unpaired, mimicking the effect of the bound ribosome and oligonucleotide. The efn2 function was used to calculate the ΔG° of RNA secondary structure based on the predicted structure using default parameters. For the bimolecular structure prediction, Bifold application of RNAstructure (version 6.2) was used[75].

**Reporting summary**. Further information on research design is available in the Nature Research Reporting Summary linked to this article.

## Data availability
The data that support this study are available from the corresponding author upon reasonable request. Source data are provided with this paper.

## Code availability
Acquisition and analysis of smFRET traces were performed using publicly available software packages from Taekjip Ha's laboratory website at Johns Hopkins University (http://ha.med.jhmi.edu/resources/) including Single software (version 0.4) and HaMMy software (version 1.0.0). Free energy change and secondary structure prediction calculations were performed with RNAstructure, which is provided free and open source under the GPL V2 license. It can be downloaded at http://rna.urmc.rochester.edu/RNAstructure.html.

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

## Acknowledgements

These studies were supported by grants from the US National Institute of Health R01GM099719, R35GM141812 (both to D.N.E.) and R01GM076485 (to D.H.M.). We thank Harry Noller, Laura Lancaster, and Gillian Rexroad for their comments on the manuscript.

## Author contributions

D.N.E. conceived the project. C.B., M.Z., D.H.M., and D.N.E. designed research. C.B. performed experiments with contributions from I.N. and H.W. M.Z. performed computational studies. C.B., M.K., D.H.M., and D.N.E. wrote the paper.

## Competing interests

The authors declare no competing interests.
