## [Peer Review File · Nature Communications]

Specific length and structure rather than high thermodynamic stability enable regulatory mRNA stem-loops to pause translationReviewers' Comments:

Reviewer #1:

Remarks to the Author:

Specific length and structure rather than high thermodynamic stability enable regulatory mRNA stem-loops to pause translation. Chen Bao, Mingyi Zhu, Inna Nykonchuk, Hironao Wakabayashi, David H. Mathews and Dmitri N. Ermolenko

This study was conducted to challenge a common assumption that thermostability of secondary RNA structures and specifically the thermostability of the three base pairs positioned at the mRNA tunnel are responsible for -1 programmed ribosomal frameshifting and ribosome pausing. To investigate this, the authors designed sequence and structure mutants of frameshift-inducing stem-loops to examine how changes in both local thermostability and structure impact translation elongation. Their study utilizes a previously established filter-binding assay to determine the effects of these mutations on tRNA A-site binding. The main conclusion is that the local thermodynamic stability of the three base pairs positioned at the mRNA tunnel did not affect A-site binding. In addition, the authors conclude that the length of the stem and the presence of the loop are important for the inhibition of A-site binding. To examine the effects of mutations on ribosomal translocation, the authors used a single molecule translation assay to measure the time it takes for ribosomes to transition from a rotated state to a stable nonrotated state, indicative of translocation. The major takeaway from these experiments is that local thermostability has a moderate effect on translocation inhibition, while stem length and presence of the loop play a key role in inhibiting translocation. This is a nice mechanistic study, but the authors need to address the following issues.

Major concerns:

1. One caveat of this study is that the authors use "specific length" to describe what causes the inability of tRNA to bind the A-site and inhibition of translocation. However, the authors only truncated, not elongated the stem of the HIV FSS. Their data shows that stem length plays a role in the HIV FSS mediated translation pausing and another study has shown that deviating from specific stem length in Simian retrovirus type-1 gag-pro transcript FSS reduce the efficiency of frameshifting, however seeing the reduction in frameshifting with an elongated stem is needed to justify their conclusions.
2. The authors need to demonstrate the dnaX construct with the truncation mutation and extended loop to see if the trend holds up in the other FSS they have tested.
3. The number of events used in the histogram analysis from their smFRET measurements. A few of their experiments have an N of 200, while others have an N closer to 100. Below I have included more specific comments.

Minor concerns:

1. Line 399-400, "altering length and sequence of the HIV FSS does not affect FSS induced inhibition of translocation." I believe this should be clarified to say altering length of sequence of the loop of HIV FSS...otherwise it contradicts the idea that specific length plays a role in the inhibition of translation.
2. Line 101, "at transcriptome-wide level," consider "on a transcriptome-wide level"
3. In the first paragraph of the results section consider elaboration on the gag-pol transcript of HIV. For dnaX you explain that the slip causes the γ transcript, but you only explain that the -1 PRF causes the gag pol transcript not what the normal transcript is.
4. Line 254, "cotrast" should read contrast.

Reviewer #2:

Remarks to the Author:

This manuscript presents data showing that pausing of ribosome translation is determined by the details of the mRNA stem-loop structures at the mRNA entry channel, rather than the thermodynamic stability of the structures. This conclusion is made based on analysis of two models of ribosome programmed -1 frame shifting, where the physical barrier of a stem-loop structure at the mRNA entry channel is a requirement that induces the ribosome to shift to the -1 frame at a slippery sequence.

However, the scope of the study is shallow and the data are not convincing.

Main points:

1. While the study is based on models of ribosome programmed -1 frameshifting, there is no data on measurements of -1 frameshifting. The two models, one on the HIV -1 frameshifting stimulating structure (FSS), and the other on E. coli dnaX FSS sequence, used in the study did not even include the cognate slippery sequence of each. Lacking the measurement of -1 frameshifting, this study is detached from the real biological questions.
2. Programmed ribosome -1 frameshifting results in un-winding of just two base pairs in the FSS sequence. This possibility was not considered.
3. Programmed ribosome -1 frameshifting occurs during translocation. While the authors performed smFRET assays to measure the dwell time of translocation, there was no mention of how the dwell time of translocation is related to -1 frameshifting. For example, the question of how long does the ribosome needs to pause to induce -1 frameshifting is not addressed.
4. The authors measured the kinetics of A-site tRNA binding as an indicator of ribosome pausing. How is this ribosome pausing due to slow kinetics of A-site tRNA binding related to -1 frameshifting?
5. The authors mentioned that the kinetics of A-site tRNA binding in an mRNA lacking a stem-loop structure is too fast to monitor using the filter binding assay. However, without this control, it is difficult to assess the sensitivity of the assay. For example, is the observed increase from 0.2 to 0.9 per min significant from one stem-loop structure to another? How many folds of change are necessary to have the confidence that it is a "significant" change?
6. All of the thermodynamic analysis of the stem-loop structures was based on prediction from a software. There was no direct assessment of the stability of each structure.
7. The authors claimed that the stability of the stem-loop structure was not important for ribosome stalling, but that the length of the stem-loop structure was. This claim is not sound. They needed to generate stem-loop structures of varying lengths that maintain the same stability as the control. The only data on the length variation is the HIV truncated FSS that has lost stability. There was no data on a shorter stem-loop that maintains the same stability as the control.
8. The data were presented mostly for the HIV model, and only selected sets were presented for the dnaX model. This raises the question of whether the conclusions made based on the HIV model are generalizable to the dnaX model. For example, the loop has no effect on the HIV model, but an effect on the dnaX model. The meaning of this difference is not clear?
9. Their work showed that destabilizing the bottom of the dnaX sequence has little effect, in contrast to what has been shown by Atkins and colleagues. The authors suggested that the discrepancy may be due to differences in model mRNA lengths. This discrepancy can be easily resolved by more experiments by the authors, but it was not done.
10. Overall, this work does not present a conceptual advance that is expected for this journal.

Minor points:

1. The stated Figure 1 that is supposed to show the actual -1 frameshifting sequence and the stem-loop structure of each model is not found.
2. While the experimental measurements for A-site binding have error bars, none for the

measurements for translocation has error bars (Figure 5, Figure 6). It is not known how many particles were analyzed.

3. The authors stated that the dwell time between the injection and the transition from R to the NR conformations reflects the translation rate. This is not accurate technically. The dwell time does not mean the rate of translocation.

Reviewer #3:

Remarks to the Author:

Protein synthesis by ribosomes can be regulated through the elongation rate and pausing during translation. One example is from the -1 programmed ribosomal frameshifting (-1 PRF), in which the frameshift stimulating sequence (FSS) of mRNA forms a specific structure to stall and stimulate the ribosome to shift its reading frame. In this report, the authors have studied what features of two typical FSSs (from HIV and *E. coli*; both are hairpin structures) are important in causing translational pausing. They measured two key parameters to assess the pausing: the rate of tRNA binding to the ribosome and the rate of ribosomal translocation to the next codon. From a series of studies, the authors conclude that an FSS hairpin having a specific stem length and a loop, instead of a high-stability structure, is important to slow down the translating ribosome. Here, the authors illustrate important mRNA structural features in regulating translation elongation, which may also be implicated in -1 PRF. Overall, this manuscript is well written and presentation of the data is mostly clear. Below are some points that need to be further clarified.

Although this manuscript is mainly to study ribosomal pausing, the mRNA structures used are typical -1 PRF stimulators (FSSs). Whether and how the FSS features identified in this study are correlated to -1 PRF should be discussed. In addition, the authors conclude that the total and the local stability of FSS moderately influence ribosome pausing. However, ref #34 showed that the frameshifting efficiency was a 5-fold increase (4.6% to 23.2%) when the stability of the first 3 bp of the HIV-1 FSS was increased, and the frameshifting efficiency was positively correlated with the stability of the first 3 bp of the FSS. Thus, frameshifting efficiency does not seem to be correlated with ribosomal pausing from this perspective.

Ref #32 showed that the total stability of the dnaX FSS (and mutants) was positively correlated with the frameshifting efficiency. The authors explain that the discrepancy is likely due to the formation of alternative structures that compete with the dnaX FSS hairpin (Supplementary Fig. 6). However, this illustrated structure involves a 2775-nucleotide insertion, which is not a compelling evidence to support their argument. For an mRNA that is repeatedly translated by ribosomes, its structures are repeatedly unfolded and refolded, and thus local structures rather than long-ranged structures should be more favorable to form. The authors should at least compare with some local structures around the FSS site.

The authors emphasized a few times that both the full-length (11 bp) and the truncated (6 bp) FSSs of HIV mRNA have the same local stability (the bottom 3 bp) for ribosome unwinding (e.g., p.9, lines 178-180). However, the distance to the transition state of a hairpin during its unfolding is about half of its length. In other words, the whole 6-bp stem of the truncated FSS is likely to be completely opened when the bottom 3 bp are unwound by the ribosome. Thus, I am not sure if the authors' statement is still correct in this regard. At least the unfolding kinetics of these two hairpins should be different during ribosomal unwinding.

In Fig. 4 and Supplementary Fig. 3-5, the histograms of dwell times are mostly like a "normal distribution", instead of a typical exponential distribution. Is this due to some sort of detection limits or involving complex kinetics?

Other minor points:

p.8, line 156: "...in HIV and dnaX FSSs by 4.7 and..." The number is 4.6 in Fig. 1b.

p.11, line 209 (and a few other places): the loop sequence should be ACAA, instead of ACCA.

p.17, line 339: "...altering length and sequence of the HIV FSS does..." It should be referring to the "loop" of the HIV FSS.

p.20, line 407: "...A-site finger (Helix 38) of 23 rRNA." => 23S rRNA

p.22, line 464: What is "entropic stabilization?" This term seems self-contradictory.

p.28, line 594: "...show the distributions and median values of represents the dwell..." Delete the "represents?"

Supplementary Fig. 2: The unit of $t_{1/2}$ on the figure should be "sec", instead of "sec⁻¹". In addition, panel c does not look like a histogram. Also, given the time span (0-180 sec), using a binning size of 0.1 sec seems unusual.

We appreciate these thoughtful reviews, which helped us improve our manuscript. To address issues raised by the reviewers, we added substantial amount of new data in the revised manuscript (Fig. 1, 3, 4, 6, 7 and 9; Supplementary Fig. S1, 2, 3, 5, 6, 7 and 8; Supplementary Tables S1, 2 and 3). Significant textual changes in the revised manuscript are highlighted in yellow. Our responses to each reviewer's concerns and suggestions are provided below (in blue).

Reviewer #1 (Remarks to the Author):

Major concerns:

1. One caveat of this study is that the authors use "specific length" to describe what causes the inability of tRNA to bind the A-site and inhibition of translocation. However, the authors only truncated, not elongated the stem of the HIV FSS. Their data shows that stem length plays a role in in the HIV FSS mediated translation pausing and another study has shown that deviating from specific stem length in Simian retrovirus type-1 gag-pro transcript FSS reduce the efficiency of frameshifting, however seeing the reduction in frameshifting with an elongated stem is needed to justify their conclusions.

To address this concern, we have extended the HIV FSS by six base pairs. We found that extending the stem of the HIV FSS completely eliminates the ability of this stem-loop to inhibit A-site tRNA binding. Furthermore, the stem-extended FSS slows down ribosome translocation by three folds, while the wild-type HIV FSS inhibited translocation by 25 folds. The effect of the extended HIV FSS on translocation is similar to effects of the truncated HIV FSS as well as HIV, dnaX and m291 bimolecular helices. This threefold reduction of translocation rate is consistent with pausing due to unwinding of three mRNA basepairs. By contrast, the 25-fold reduction of translocation rate induced by the wild-type HIV FSS likely results from the formation of specific inhibitory FSS-ribosome interactions. These new results are included in Fig. 1, 4, 6 and 7 of the revised manuscript.

Because the dnaX FSS contains two bulges (unpaired nucleotides in the stem) designing mutant versions of dnaX FSS with extended and especially truncated RNA stems was problematic. For that reason, we varied the stem length in the HIV FSS only. We included this comment in the manuscript.

2. The authors need to demonstrate the dnaX construct with the truncation mutation and extended loop to see if the trend holds up in the other FSS they have tested.

We extended the loop in the dnaX FSS from 5 to 10 nucleotides. Similar to the results obtained with the HIV FSS, the extension of the loop did not significantly affect the FSS-induced inhibition of A-site tRNA binding or translocation. These new results are included in Fig. 1, 3 and 6 of the revised manuscript.

As mentioned above, because the dnaX FSS contains two bulges (unpaired nucleotides in the stem), designing mutant versions of dnaX FSS with a truncated RNA stem while keeping the FSS sufficiently stable was not feasible. For this reason, we did not attempt experiments with the truncated dnaX FSS.

3. The number of events used in the histogram analysis from their smFRET measurements. A few of their experiments have an N of 200, while others have an N closer to 100. Below I have included more specific comments.

In addition to median translocation times shown in Fig. 6, we included Supplementary Fig.5 that shows translocation times averaged from 5 to 20 independent experiments (~10-30 traces per each experiment) and errors bars corresponding to standard deviations (thin error bars) and standard errors (thick error bars) of the mean, respectively. These data demonstrate reproducibility and self-consistency in our smFRET experiments.

Minor concerns:

1. Line 399-400, “altering length and sequence of the HIV FSS does not affect FSS induced inhibition of translocation.” I believe this should be clarified to say altering length of sequence of the loop of HIV FSS...otherwise it contradicts the idea that specific length plays a role in the inhibition of translation.

We made this correction and inserted the word “loop”.

2. Line 101, “at transcriptome-wide level,” consider “on a transcriptome-wide level”

We made this correction.

3. In the first paragraph of the results section consider elaboration on the gag-pol transcript of HIV. For dnaX you explain that the slip causes the γ transcript, but you only explain that the -1 PRF causes the gag pol transcript not what the normal transcript is.

We elaborated on HIV PRF: “While the Gag polyprotein containing HIV structural proteins is produced in the absence of frameshifting, the 12 basepair-long RNA hairpin in combination with the slippery sequence UUUUUUA induces -1 PRF with 5-10% efficiency. This frameshifting yields the Gag-Pol polyprotein, which contains reverse transcriptase, RNase H, integrase and HIV protease”.

4. Line 254, “cotrast” should read contrast.

We made this correction.

Reviewer #2 (Remarks to the Author):

Main points:

1. While the study is based on models of ribosome programmed -1 frameshifting, there is no data on measurements of -1 frameshifting. The two models, one on the HIV -1 frameshifting stimulating structure (FSS), and the other on E. coli dnaX FSS sequence, used in the study did not even include the cognate slippery sequence of each. Lacking the measurement of -1 frameshifting, this study is detached from the real biological questions.

In our opinion, the questions “how mRNA stem-loops regulate the rate of translation?” and “why do some elements of mRNA secondary structure slow down the ribosome much more than others?”, which are addressed in our manuscript, have much broader biological implications than the mechanism of programmed frameshifting alone. mRNA has a propensity to form extensive secondary structure, and thus mRNA stem-loops likely affect translation of many if not all mRNAs. Finding why some stem-loops slow down the ribosome more than others is important for interpreting ribosome profiling data, discovering new regulatory stem-loops and designing mRNAs for biotechnology (including mRNA vaccines). We added this statement to Introduction.

Because ribosome frameshifting partially alleviates FSS-induced ribosome pausing and changes the spacing between the P-site codon and the FSS, we studied FSS-induced pausing decoupled from ribosome frameshifting, i.e. in the absence of the slippery sequence. This approach (replacing slippery codons with non-slippery ones to study stem-loop-induced pausing) has been successfully employed by a number of research groups¹⁻⁴.

We have also investigated the FSS effects on tRNA binding and translocation during frameshifting (i.e. in the presence of the slippery sequence) in our previous publication¹. Furthermore, several other groups (Rodnina, Puglisi, Gonzalez/Tinoco) also used single-molecule and biochemical experiments to research the role of the FSS in ribosome frameshifting. From our point of view, the novelty of the current manuscript stems from the focus on structural properties of mRNA stem-loops important for inducing ribosome pausing.

Although our manuscript is focused on FSS-induced ribosome pausing alone (and not frameshifting), we added to the revised manuscript new measurements of frameshifting efficiency. Our data show that mutations destabilizing the bottom basepairs of the dnaX FSS have a lesser effect on the frameshifting rate than replacing the dnaX FSS with a bi-molecular helix identical to the stem of dnaX FSS. That bi-molecular helix nearly completely diminished the ability of the FSS to stimulate frameshifting. These results are consistent with our biochemical and smFRET data obtained in the absence of the slippery sequence supporting the idea that the extent of ribosome pausing correlates with the ability of the FSS to stimulate ribosome frameshifting.

2. Programmed ribosome -1 frameshifting results in un-winding of just two base pairs in the FSS sequence. This possibility was not considered.

We added the following to Discussion: “Differences between effects of destabilizing mutations in FSSs on ribosome pausing and frameshifting might indicate that the duration of FSS-induced ribosome pausing weakly correlates with frameshifting efficiency. Furthermore, during translocation coupled with -1 PRF, two instead of three bottom basepairs of the FSS are unwound, further complicating the comparison of ribosome pausing observed in the context of non-slippery codons with frameshifting. Besides, in contrast to short model mRNAs used in our experiments, FSS sequences in long mRNA encoding frameshifting reporters may form several alternative structures by forming basepairs with the rest of mRNA, which compete with FSS basepairs. It is possible that the mutations designed to destabilize the FSS resulted in alternative secondary structures, which complicate the analysis of the data. By contrast, stabilizing mutations in the FSS might increase frameshifting efficiency by enlarging the fraction of mRNA, which contains properly folded FSS, rather than by hindering FSS unwinding by the ribosome”.

3. Programmed ribosome -1 frameshifting occurs during translocation. While the authors performed smFRET assays to measure the dwell time of translocation, there was no mention of how the dwell time of translocation is related to -1 frameshifting. For example, the question of how long does the ribosome needs to pause to induce -1 frameshifting is not addressed.

In the revised manuscript, we expanded discussion of correlation between ribosome pausing and frameshifting (please see our response to the previous comment). Our experiments indicate that a bimolecular helix that is identical to the stem of dnaX FSS barely stimulates frameshifting above levels observed in the absence of the FSS (Fig. 9 of revised manuscript). Because the same bimolecular helix produces only 3-fold reduction in translocation rate relative to Δ FSS control, longer duration ribosomal pauses are needed to substantially stimulate ribosome frameshifting.

4. The authors measured the kinetics of A-site tRNA binding as an indicator of ribosome pausing. How is this ribosome pausing due to slow kinetics of A-site tRNA binding related to -1 frameshifting?

Multiple lines of evidence indicate that -1 PRF can occur through three different pathways: (i) slippage of the single P-site tRNA when the A site remains vacant, (ii) frameshifting of both A-site and P-site tRNAs during aa-tRNA accommodation to the A site, and (iii) slippage of A- and P-site tRNAs during translocation⁵. A number of studies showed that -1 PRF on both *dnaX* and HIV mRNAs involves P-site tRNA slippage and frameshifting during translocation of two tRNAs, i.e. pathways (i) and (iii)⁶⁻¹¹. Studies by the Rodnina lab demonstrated that partitioning between these pathways is modulated by tRNA abundance^{6,7,12}. Our observation that frameshift-inducing hairpins inhibit both A-site tRNA binding and mRNA translocation are consistent with single tRNA slippage (i) and translocation (iii) frameshifting pathways, respectively.

5. The authors mentioned that the kinetics of A-site tRNA binding in an mRNA lacking a stem-loop structure is too fast to monitor using the filter binding assay. However, without this control, it is difficult to assess the sensitivity of the assay. For example, is the observed increase from 0.2 to 0.9 per min significant from one

stem-loop structure to another? How many folds of change are necessary to have the confidence that it is a “significant” change?

Fig. 7e of the revised manuscript indicates that under experimental conditions used in this work, in the absence of the HIV FSS, Tyr^rtRNA binds to the A site at the rate of $\sim 0.3 \text{ s}^{-1}$ (18 min^{-1}). Hence, the wild-type and truncated FSS HIV inhibit the rate of A-site binding ~ 100 and 20 fold, respectively. Supplementary table S3 shows p values indicating that truncation of the HIV FSS and extending the loop in HIV FSS produce statistically significant changes in the rate of A-site tRNA binding.

6. All of the thermodynamic analysis of the stem-loop structures was based on prediction from a software. There was no direct assessment of the stability of each structure.

Nearest neighbor estimation of helix stabilities is well established as accurate¹³. Given that most comparisons in Figure 1 are between constructs that differ by canonical base pairing, we are confident that the relative stabilities are correct. We modified Figure 1 to provide error estimates for each calculated stability, using recent quantification of the confidence of nearest neighbor parameters by propagating the experimental errors through the derivation of parameters^{14,15}. The error estimates are smaller than the calculated $\Delta\Delta G^\circ$ s.

7. The authors claimed that the stability of the stem-loop structure was not important for ribosome stalling, but that the length of the stem-loop structure was. This claim is not sound. They needed to generate stem-loop structures of varying lengths that maintain the same stability as the control. The only data on the length variation is the HIV truncated FSS that has lost stability. There was no data on a shorter stem-loop that maintains the same stability as the control.

Thermodynamic stability of RNA helices is a function of helix length (number of basepairs)¹³. It is impossible to truncate a stem-loop without lowering stem-loop stability especially when the original stem-loop is GC rich (as HIV FSS). Additionally, if the stems were made much shorter, it would be hard to design sequences that would form the specific stem with high probability. Short stems have fewer unique pairs, which are needed for specific structures.

8. The data were presented mostly for the HIV model, and only selected sets were presented for the dnaX model. This raises the question of whether the conclusions made based on the HIV model are generalizable to the dnaX model. For example, the loop has no effect on the HIV model, but an effect on the dnaX model. The meaning of this difference is not clear?

We extended the loop in dnaX FSS from 5 to 10 nucleotides (Fig. 1). Similar to HIV FSS, the loop extension did not have affect the ability of dnaX FSS to inhibit both A-site binding and translocation (Fig. 3). Because the dnaX FSS contains two bulges (unpaired nucleotides in the stem) designing mutant versions of dnaX FSS with extended and especially truncated RNA stems was problematic. For that reason, we varied the stem length in the HIV FSS only. We included this comment in the manuscript.

9. Their work showed that destabilizing the bottom of the dnaX sequence has little effect, in contrast to what has been shown by Atkins and colleagues. The authors suggested that the discrepancy may be due to differences in model mRNA lengths. This discrepancy can be easily resolved by more experiments by the authors, but it was not done.

We revised our text explaining why the experiments by Atkins and colleagues are complicated by competing secondary structures. We are not sure which experiments the reviewer had in mind. Using short model dnaX mRNAs, we measured frameshifting efficiency in the presence of the dnaX bimolecular helix, as well as destabilized and wild-type dnaX FSSs. We found that consistent with measurements of ribosome pausing;

destabilizing the bottom basepairs of dnaX FSS decreases frameshifting efficiency while replacing the wild-type FSS with a bi-molecular helix identical to the stem of dnaX FSS diminishes frameshifting close to the levels observed in the absence of the FSS. We also added more detailed discussion of correlation between frameshifting and ribosome pausing (please see above).

10. Overall, this work does not present a conceptual advance that is expected for this journal.

We respectfully disagree. Finding that specific length and structure enable some mRNA stem-loops to slow down the ribosome at least an order of magnitude more than other elements of mRNA secondary structure is a significant advance in understanding of translation regulation that has a number of broad implications.

Minor points:

1. The stated Figure 1 that is supposed to show the actual -1 frameshifting sequence and the stem-loop structure of each model is not found.

Figure 1 was present in the originally submitted manuscript. The reviewer's concern is unclear to us.

2. While the experimental measurements for A-site binding have error bars, none for the measurements for translocation has error bars (Figure 5, Figure 6). It is not known how many particles were analyzed.

The number of traces analyzed for each mRNA is indicated on Fig. 5, Fig. 7 and Fig. 8 (original Fig. 6). Likewise, the number of traces used to calculate the mean translocation time for each mRNA (Fig. 6, original Fig. 5) is shown in Supplementary. Fig. S4, 6 and 7. We also added average translocation times and standard error of the mean calculated from 5 to 20 independent experiments (10-30 traces in each experiment) in Supplementary Fig. S5.

3. The authors stated that the dwell time between the injection and the transition from R to the NR conformations reflects the translation rate. This is not accurate technically. The dwell time does not mean the rate of translocation.

Dwell time is inversely proportional to the rate. Hence, to be more technically accurate, we used phrasing "dwell time reflects the rate".

Reviewer #3 (Remarks to the Author):

Protein synthesis by ribosomes can be regulated through the elongation rate and pausing during translation. One example is from the -1 programmed ribosomal frameshifting (-1 PRF), in which the frameshift stimulating sequence (FSS) of mRNA forms a specific structure to stall and stimulate the ribosome to shift its reading frame. In this report, the authors have studied what features of two typical FSSs (from HIV and E. coli; both are hairpin structures) are important in causing translational pausing. They measured two key parameters to assess the pausing: the rate of tRNA binding to the ribosome and the rate of ribosomal translocation to the next codon. From a series of studies, the authors conclude that an FSS hairpin having a specific stem length and a loop, instead of a high-stability structure, is important to slow down the translating ribosome. Here, the authors illustrate important mRNA structural features in regulating translation elongation, which may also be implicated in -1 PRF. Overall, this manuscript is well written and presentation of the data is mostly clear. Below are some points that need to be further clarified.

Although this manuscript is mainly to study ribosomal pausing, the mRNA structures used are typical -1 PRF stimulators (FSSs). Whether and how the FSS features identified in this study are correlated to -1 PRF should

be discussed. In addition, the authors conclude that the total and the local stability of FSS moderately influence ribosome pausing. However, ref #34 showed that the frameshifting efficiency was a 5-fold increase (4.6% to 23.2%) when the stability of the first 3 bp of the HIV-1 FSS was increased, and the frameshifting efficiency was positively correlated with the stability of the first 3 bp of the FSS. Thus, frameshifting efficiency does not seem to correlated with ribosomal pausing from this perspective.

In response to this concern, we measured frameshifting efficiency in dnaX constructs including the slippery sequence (Fig. 1 and Fig. 9). Using short model dnaX mRNAs, we measured frameshifting efficiency in the presence of the dnaX bimolecular helix, as well as destabilized and wild-type dnaX FSSs. We found that consistent with measurements of ribosome pausing; destabilizing the bottom basepairs of dnaX FSS decreases frameshifting efficiency while replacing the wild-type FSS with a bi-molecular helix identical to the stem of dnaX FSS diminishes frameshifting close to the levels observed in the absence of the FSS. We also significantly expanded discussion of correlation between ribosome pausing and frameshifting efficiency. In particular, we acknowledge the possibility that frameshifting efficiency weakly correlates with duration of ribosome pauses.

Ref #32 showed that the total stability of the dnaX FSS (and mutants) was positively correlated with the frameshifting efficiency. The authors explain that the discrepancy is likely due to the formation of alternative structures that compete with the dnaX FSS hairpin (Supplementary Fig. 6). However, this illustrated structure involves a 2775-nucleotide insertion, which is not a compelling evidence to support their argument. For an mRNA that is repeatedly translated by ribosomes, its structures are repeatedly unfolded and refolded, and thus local structures rather than long-ranged structures should be more favorable to form. The authors should at least compare with some local structures around the FSS site.

We removed original Supplementary Fig. 6 and revised discussion of correlation between frameshifting efficiency and ribosome pausing as follows:

“We find that the total thermodynamic stability of the FSSs and the local stability of the basepairs adjacent to the entry of mRNA tunnel moderately influence ribosome pausing. Hence, instead of merely acting as a physical barrier for the translating ribosome, the FSS likely hinder protein synthesis by forming specific interactions with the ribosome. Consistent with this idea, the lack of correlation between total thermodynamic stability of the FSS and the efficiency of frameshifting has been observed in several studies, including our own experiments (Fig. 9). In addition, significant destabilization of the bottom three basepairs of the HIV FSS has been shown to have no effect on the efficiency of frameshifting.

By contrast, stabilization of the bottom basepairs of the HIV FSS has been shown to substantially increase frameshifting efficiency. Furthermore, positive correlation between frameshifting efficiency and the overall stability of dnaX FSS have been demonstrated by Atkins and colleagues. Differences between effects of destabilizing mutations in FSSs on ribosome pausing and frameshifting might indicate that the duration of FSS-induced ribosome pausing weakly correlates with frameshifting efficiency. Furthermore, during translocation coupled with -1 PRF, two instead of three bottom basepairs of the FSS are unwound, further complicating the comparison of ribosome pausing observed in the context of non-slippery codons with frameshifting. Besides, in contrast to short model mRNAs used in our experiments, FSS sequences in long mRNA encoding frameshifting reporters may form several alternative structures by forming basepairs with the rest of mRNA, which compete with FSS basepairs. It is possible that the mutations designed to destabilize the FSS resulted in alternative secondary structures, which complicate the analysis of the data. By contrast, stabilizing mutations in the FSS might increase frameshifting efficiency by enlarging the fraction of mRNA, which contains properly folded FSS, rather than by hindering FSS unwinding by the ribosome”.

The authors emphasized a few times that both the full-length (11 bp) and the truncated (6 bp) FSSs of HIV

mRNA have the same local stability (the bottom 3 bp) for ribosome unwinding (e.g., p.9, lines 178-180). However, the distance to the transition state of a hairpin during its unfolding is about half of its length. In other words, the whole 6-bp stem of the truncated FSS is likely to be completely opened when the bottom 3 bp are unwound by the ribosome. Thus, I am not sure if the authors' statement is still correct in this regard. At least the unfolding kinetics of these two hairpins should be different during ribosomal unwinding.

We respectfully disagree that the transition state of a hairpin stem-loop would necessarily be half the length of the stem. For an isolated hairpin stem-loop that is partially formed, a folding free energy < 0 kcal/mol would mean that the partial stem is stable, and therefore the rate of forming more pairs (and lowering the folding free energy change) would exceed the rate of removing pairs (and increasing the folding free energy change). This follows from kinetic models with secondary structure as their models^{16,17}. For even the truncated (6 bp) HIV FSS, with only the top 3 pairs, the estimated folding free energy change is -2.6 ± 0.2 kcal/mol. Therefore, we conclude that for all the stem-loop structures, the unwinding of the first 3 pairs would not destabilize the hairpin enough that the remaining structure would denature.

In Fig. 4 and Supplementary Fig. 3-5, the histograms of dwell times are mostly like a "normal distribution", instead of a typical exponential distribution. Is this due to some sort of detection limits or involving complex kinetics?

Translocation kinetics in the presence of the stem-loop is likely complex because ribosome translocation requires unwinding of three basepairs and, possibly, rearrangement of the stem-loop relative to the ribosome from the conformation inhibiting unwinding to the conformation conducive to unwinding. These steps preceding translocation produce a noticeable pause in dwell time distributions.

Other minor points:

p.8, line 156: "...in HIV and dnaX FSSs by 4.7 and..." The number is 4.6 in Fig. 1b.

We've corrected that error.

p.11, line 209 (and a few other places): the loop sequence should be ACAA, instead of ACCA.

We've corrected that error.

p.17, line 339: "...altering length and sequence of the HIV FSS does..." It should be referring to the "loop" of the HIV FSS.

We've corrected that error.

p.20, line 407: "...A-site finger (Helix 38) of 23 rRNA." => 23S rRNA

We wanted to indicate the specific location along 23S rRNA. For that reason, we've left original wording unchanged.

p.22, line 464: What is "entropic stabilization?" This term seems self-contradictory.

We've changed the wording of this sentence: "Alternatively, the short (11 nucleotide) spacer restricts the FSS dynamics and thus decreases entropic penalty for forming FSS-ribosome interactions that inhibit translocation".

p.28, line 594: "...show the distributions and median values of represents the dwell..." Delete the "represents?"

We've deleted "represents".

Supplementary Fig. 2: The unit of $t_{1/2}$ on the figure should be “sec”, instead of “sec⁻¹”. In addition, panel c does not look like a histogram. Also, given the time span (0-180 sec), using a binning size of 0.1 sec seems unusual.

We corrected units for $t_{1/2}$. Data in panel c were not binned in that each bin corresponded to the number of frames in the movie acquired with 0.1 s resolution. We have replotted the histogram of panel c with 1 s binning size (i.e. same binning as in panels a and b).

References:

1. Bao, C. et al. mRNA stem-loops can pause the ribosome by hindering A-site tRNA binding. *Elife* **9**(2020).
2. Zhang, Y., Hong, S., Ruangprasert, A., Skiniotis, G. & Dunham, C.M. Alternative Mode of E-Site tRNA Binding in the Presence of a Downstream mRNA Stem Loop at the Entrance Channel. *Structure* **26**, 437-445 e3 (2018).
3. Qin, P., Yu, D., Zuo, X. & Cornish, P.V. Structured mRNA induces the ribosome into a hyper-rotated state. *EMBO Rep* **15**, 185-90 (2014).
4. Yan, S., Wen, J.D., Bustamante, C. & Tinoco, I., Jr. Ribosome excursions during mRNA translocation mediate broad branching of frameshift pathways. *Cell* **160**, 870-81 (2015).
5. Dinman, J.D. Mechanisms and implications of programmed translational frameshifting. *Wiley Interdiscip Rev RNA* **3**, 661-73 (2012).
6. Caliskan, N. et al. Conditional Switch between Frameshifting Regimes upon Translation of dnaX mRNA. *Mol Cell* **66**, 558-567 e4 (2017).
7. Korniy, N. et al. Modulation of HIV-1 Gag/Gag-Pol frameshifting by tRNA abundance. *Nucleic Acids Res* **47**, 5210-5222 (2019).
8. Jacks, T. et al. Characterization of ribosomal frameshifting in HIV-1 gag-pol expression. *Nature* **331**, 280-3 (1988).
9. Brunelle, M.N., Payant, C., Lemay, G. & Brakier-Gingras, L. Expression of the human immunodeficiency virus frameshift signal in a bacterial cell-free system: influence of an interaction between the ribosome and a stem-loop structure downstream from the slippery site. *Nucleic Acids Res* **27**, 4783-91 (1999).
10. Leger, M., Dulude, D., Steinberg, S.V. & Brakier-Gingras, L. The three transfer RNAs occupying the A, P and E sites on the ribosome are involved in viral programmed -1 ribosomal frameshift. *Nucleic Acids Res* **35**, 5581-92 (2007).
11. Yelverton, E., Lindsley, D., Yamauchi, P. & Gallant, J.A. The function of a ribosomal frameshifting signal from human immunodeficiency virus-1 in Escherichia coli. *Mol Microbiol* **11**, 303-13 (1994).
12. Korniy, N., Samatova, E., Anokhina, M.M., Peske, F. & Rodnina, M.V. Mechanisms and biomedical implications of -1 programmed ribosome frameshifting on viral and bacterial mRNAs. *FEBS Lett* **593**, 1468-1482 (2019).
13. Xia, T. et al. Thermodynamic parameters for an expanded nearest-neighbor model for formation of RNA duplexes with Watson-Crick base pairs. *Biochemistry* **37**, 14719-35 (1998).
14. Zuber, J., Cabral, B.J., McFadyen, I., Mauger, D.M. & Mathews, D.H. Analysis of RNA nearest neighbor parameters reveals interdependencies and quantifies the uncertainty in RNA secondary structure prediction. *RNA* **24**, 1568-1582 (2018).
15. Zuber, J. & Mathews, D.H. Estimating uncertainty in predicted folding free energy changes of RNA secondary structures. *RNA* **25**, 747-754 (2019).
16. Flamm, C., Fontana, W., Hofacker, I.L. & Schuster, P. RNA folding at elementary step resolution. *RNA* **6**, 325-38 (2000).
17. Zolaktaf, S. et al. Inferring Parameters for an Elementary Step Model of DNA Structure Kinetics with Locally Context-Dependent Arrhenius Rates. *DNA Computing and Molecular Programming - 23rd International Conference, DNA 23, Austin, TX, USA, September 24-28, 2017, Proceedings* **10467**, 172-187 (2017).

Reviewers' Comments:

Reviewer #2:

Remarks to the Author:

Reviewer #2:

Most of the concerns raised by Reviewer #2 have been adequately addressed. However, minor revisions should be made to address the three remaining points below.

Point #2: While the revised Discussion addressed multiple points, the text was very scattered and it is difficult to have the "take-home" message. The authors should add one concluding remark at the end of the paragraph that summarizes all of the points here and provide a clear message to the readers.

Point #6: Because the thermodynamic free energy changes were completely based on calculation, and no experimental measurements provided, the authors should not be completely "confident" that their calculated stabilities were accurate. The statement in the text should be tuned down and should include the acknowledgement that no actual measurements were done.

Point #7: the authors should include the discussion presented in the point-by-point response into the text in the Discussion section.

Reviewer #3:

Remarks to the Author:

My concerns have been mostly addressed in the revised manuscript, except for the following minor points. Other than that, I do not have further comments.

Lines 476-477: "In addition, significant destabilization of the bottom three basepairs of the HIV FSS has been shown to have no effect on the efficiency of frameshifting³⁴." Here, this reference (#34) appears not to be an appropriate citation, as its main conclusion is that the frameshifting efficiency is strongly correlated with the thermodynamic stability of the first 3-4 bp of the FSS hairpin (as similarly pointed out by the authors in lines 478-479).

Lines 508-509: "In the HIV FSS-ribosome structure, the FSS loop contacts the A-site finger (Helix 38) of 23 rRNA." I still do not follow the authors' point in their responses. What does the number "23" exactly refer to? Doesn't it indicate the "23S" rRNA?

Lines 754-755: "... show the distributions and median values the dwell time ..." Here, an "of" preceding "the dwell time" was deleted.

Our responses to each reviewer's concerns and suggestions are provided below (in blue).

Reviewer #2:

Most of the concerns raised by Reviewer #2 have been adequately addressed. However, minor revisions should be made to address the three remaining points below.

Point #2: While the revised Discussion addressed multiple points, the text was very scattered and it is difficult to have the "take-home" message. The authors should add one concluding remark at the end of the paragraph that summarizes all of the points here and provide a clear message to the readers.

We revised the original paragraph in the Discussion in order to provide a clear message.

Original wording: "Differences between effects of destabilizing mutations in FSSs on ribosome pausing and frameshifting might indicate that the duration of FSS-induced ribosome pausing weakly correlates with frameshifting efficiency. Furthermore, during translocation coupled with -1 PRF, two instead of three bottom basepairs of the FSS are unwound, further complicating the comparison of ribosome pausing observed in the context of non-slippery codons with frameshifting. Besides, in contrast to short model mRNAs used in our experiments, FSS sequences in long mRNA encoding frameshifting reporters may form several alternative structures by forming basepairs with the rest of mRNA, which compete with FSS basepairs. It is possible that the mutations designed to destabilize the FSS resulted in alternative secondary structures, which complicate the analysis of the data. By contrast, stabilizing mutations in the FSS might increase frameshifting efficiency by enlarging the fraction of mRNA, which contains properly folded FSS, rather than by hindering FSS unwinding by the ribosome".

New wording: "Several factors, however, complicate the comparison of ribosome pausing observed in the context of non-slippery codons in our work with previously reported measurements of frameshifting. First, during translocation coupled with -1 PRF, two instead of three bottom basepairs of the FSS are unwound. Second, in contrast to short model mRNAs used in our experiments, FSS sequences in long mRNA encoding frameshifting reporters may form several alternative structures by forming basepairs with the rest of mRNA, which compete with FSS basepairs. It is possible that the mutations designed to destabilize the FSS resulted in alternative secondary structures, which complicate the analysis of the data. By contrast, stabilizing mutations in the FSS might increase frameshifting efficiency by enlarging the fraction of mRNA, which contains properly folded FSS, rather than by hindering FSS unwinding by the ribosome. Although the duration of FSS-induced ribosome pausing might weakly correlate with frameshifting efficiency, it is more likely that apparent discrepancies between measurements of ribosome pausing and frameshifting are due to formation of alternative secondary structures in long reporter mRNAs".

Point #6: Because the thermodynamic free energy changes were completely based on calculation, and no experimental measurements provided, the authors should not be completely "confident" that their calculated stabilities were accurate. The statement in the text should be tuned down and should include the acknowledgement that no actual measurements were done.

We added following sentences to the first paragraph of Discussion to clarify that the free energy changes are calculations: "In our work, we used nearest neighbor estimation of helix stabilities, which is well established as accurate⁴⁷. Using recent quantification of the confidence of nearest neighbor parameters by propagating the experimental errors through the derivation of parameters^{61,62}, we found that the error estimates are smaller than the calculated $\Delta\Delta G^\circ$ s (Fig.1)".

Point #7: the authors should include the discussion presented in the point-by-point response into the text in the Discussion section.

We added a sentence to the description of truncated HIV FSS to summarize the most important parts of the response: "Because thermodynamic stability of RNA helices is a function of helix length (number of basepairs),

it is impossible to truncate a stem-loop without lowering overall stem-loop stability". Because *Nature Communications* publishes responses to reviewers' comments, the complete discussion to point #7 will also be available.

Reviewer #3 (Remarks to the Author):

My concerns have been mostly addressed in the revised manuscript, except for the following minor points. Other than that, I do not have further comments.

Lines 476-477: "In addition, significant destabilization of the bottom three basepairs of the HIV FSS has been shown to have no effect on the efficiency of frameshifting³⁴." Here, this reference (#34) appears not to be an appropriate citation, as its main conclusion is that the frameshifting efficiency is strongly correlated with the thermodynamic stability of the first 3-4 bp of the FSS hairpin (as similarly pointed out by the authors in lines 478-479).

The main conclusion of ref 34 is indeed that the frameshifting efficiency is strongly correlated with the thermodynamic stability of the first 3-4 bp of the FSS hairpin. This conclusion is predominately drawn from data indicating that stabilization of the bottom basepairs in the FSS enhance frameshifting efficiency. However, data presented in this paper also indicate that mutations, which destabilize the bottom basepairs, had negligible effect on frameshifting efficiency. Here we refer to this subset of data presented in ref 34.

Lines 508-509: "In the HIV FSS-ribosome structure, the FSS loop contacts the A-site finger (Helix 38) of 23 rRNA." I still do not follow the authors' point in their responses. What does the number "23" exactly refer to? Doesn't it indicate the "23S" rRNA?

We misunderstood the original point of the reviewer. We corrected the typo (missing S in 23S rRNA).

Lines 754-755: "... show the distributions and median values the dwell time ..." Here, an "of" preceding "the dwell time" was deleted.

We corrected that typo and inserted "of".